# Fast Generation, Forecasting, and Imputation of Multivariate Irregular Time Series with OUFlow

## Abstract

We propose OUFlow, a general-purpose time-series generative model that robustly handles irregular sampling and generates sequences at arbitrary time points. OUFlow integrates latent dynamics governed by a mixture of Ornstein-Uhlenbeck processes with expressive target distributions via normalizing flows. Leveraging our analytically derived, efficiently computable likelihoods and posteriors for high-dimensional time series, OUFlow supports unconditional time-series generation, probabilistic forecasting, and imputation from partial observations within a unified model after a single training phase. It also enables explicit likelihood evaluation (e.g., for anomaly detection), clustering via modes of the latent OU process, and, in some cases, denoising under noisy supervision. By exploiting parallelization through the scan algorithm, OUFlow attains logarithmic runtime scaling in the number of generated points, while maintaining high accuracy in all three tasks. Comprehensive experiments on both synthetic and real-world datasets demonstrate that OUFlow consistently outperforms other models capable of all three tasks, in both generation quality and computational efficiency.

## 1 Introduction

As time-series data proliferate across domains such as meteorology, finance, healthcare, and energy, they increasingly demand a broad spectrum of capabilities, including probabilistic generation, forecasting, imputation, anomaly detection, and beyond. Despite this breadth, most existing approaches remain narrowly task-specific, limiting methodological consistency, reuse, and scalability. These limitations, together with the growing use of foundation models whose training involves very large model capacity and prolonged training time, motivate a unified framework that operates across tasks, sampling schemes, and resolutions.

Generative models for time series are a natural basis for such a framework. In practice, they are typically developed for one or two tasks such as unconditional generation, forecasting, or imputation, but rarely support all three within the same model. Unconditional generation produces samples without conditioning on observations, enabling applications such as creative audio and video synthesis and privacy-preserving data release in sensitive domains including vital signs. Forecasting and imputation, in contrast, condition on partial observations to predict future trajectories or fill in missing segments. In chaotic systems such as weather and in human-influenced processes such as financial markets, deterministic prediction is infeasible. Probabilistic forecasts are essential for representing uncertainty, supporting decision-making, and enabling calibrated risk assessments.

A central technical challenge is irregular sampling. Real-world time series are often measured at inconsistent intervals due to sensor constraints, adaptive sampling, or missingness. Many models assume regular grids, yet admitting arbitrary observation times better reflects operational practice and improves downstream utility. Equally important is the ability to generate at arbitrary resolutions, which enables multi-scale synthesis, temporal super-resolution, and seamless alignment with heterogeneous downstream requirements.

Meeting these requirements typically introduces additional computational burdens as shown in Table 1. Models that support generation, forecasting, imputation, and irregular sampling frequently depend on numerical integration of differential equations to encode observations and

to simulate latent trajectories. In practice, this reliance introduces a largely fixed computational overhead that persists regardless of the number of observation or generation points, which can be a poor fit for workloads that require rapid turnaround or fine-grained resampling.

We introduce **OUFlow**, a scalable and general-purpose generative model for multivariate and irregularly sampled time series. OUFlow is a latent-variable model whose dynamics are governed by a mixture of Ornstein-Uhlenbeck processes, paired with a normalizing-flow decoder that captures nonlinear and non-Gaussian structure in real data. The OUFlow prior and posterior are available in closed form for arbitrary observation times, which enables a single trained model to support unconditional generation, forecasting, and imputation on arbitrary observation and generation grids. Importantly,

Table 1: Multivariate time-series generative models that can generate, forecast, and impute irregular time-series data, along with their time complexities for data generation. $N$ represents the number of observations, $K$ denotes the number of generated time points, and $T_{\text{enc}}$ and $T_{\text{dec}}$ refer to the number of necessary time points required for time integration during the conditioning (encoding) and generation (decoding) phases, respectively. Typically, $T_{\text{enc}}$ and $T_{\text{dec}}$ are much larger than $N$ and $K$.

| MODEL | GENERATION TIME |
|---|---|
| GPR | $O\left((N+K)^3\right)$ |
| LATENTODE (RUBANOVA ET AL., 2019) | $\gtrsim O(T_{\text{enc}} + K)$ |
| LATENTSDE (LI ET AL., 2020) | $\gtrsim O(T_{\text{enc}} + K)$ |
| CTFP (DENG ET AL., 2020) | $O(K)$ |
| LATENTCTFP (DENG ET AL., 2020) | $O(T_{\text{enc}} + K)$ |
| CLPF (DENG ET AL., 2021) | $O(T_{\text{enc}} + T_{\text{dec}})$ |
| GT-GAN (JEON ET AL., 2022) | $O(T_{\text{enc}} + T_{\text{dec}})$ |
| NCDSSM (ANSARI ET AL., 2023) | $\gtrsim O(N+K)$ |
| DSPD/CSPD (BILOŠ ET AL., 2023) | $\gtrsim O(N+K)$ |
| OUFLOW (OURS) | $O\left(N + \log(N+K)\right)$ |

OUFlow is integration-free. It leverages a scan-style algorithm for OU transitions that yields sampling with logarithmic time complexity in the generated sequence length, which improves scalability for high-resolution generation and flexible resampling.

Beyond core tasks, OUFlow offers broader applicability than conventional time-series generative models. Unlike VAE-based, GAN-based, and diffusion-based approaches, OUFlow provides explicit likelihoods for irregularly sampled sequences, supporting likelihood-based anomaly detection. The mixture structure of the latent OU processes induces a natural clustering of sequences, with posterior responsibilities yielding probabilistic cluster assignments. Moreover, by endowing the latent dynamics and decoder with distinct covariance structures, OUFlow can, under appropriate assumptions, disentangle system noise from observation noise, enabling system identification, data assimilation, and denoising.

**Our main contributions are:**

- We propose OUFlow, a general-purpose generative model that unifies unconditional generation, forecasting, and imputation for multivariate and irregularly sampled time series.

- We derive closed-form likelihoods and transition probabilities in a computationally efficient form for mixture OU latent dynamics paired with a normalizing-flow decoder, which enables efficient computation even for high-dimensional time series.

- We develop an integration-free sampling algorithm with logarithmic time complexity in the number of generated steps, which enables fast high-resolution synthesis and scalable inference.

- Through rigorous evaluations accounting for the distributional properties of generated time series, we demonstrate that OUFlow achieves higher accuracy and better computational efficiency with respect to the number of generated time steps than existing methods across a variety of tasks.

## 2 RELATED WORK

**Generative Models for Irregular Time Series**  Models capable of handling irregular time series data include those based on normalizing flows (Rezende & Mohamed, 2015), such as continuous-time flow process (CTFP), LatentCTFP (Deng et al., 2020), and continuous latent process flows

(CLPFs) (Deng et al., 2021). VAE (Kingma, 2013)-based models, including LatentODE (Rubanova et al., 2019), LatentSDE (Li et al., 2020), NCDSSM (Ansari et al., 2023), KoVAE (Naiman et al., 2024), and ACSSM (Park et al., 2025) also exist. GAN (Goodfellow et al., 2014)-based models such as GT-GAN (Jeon et al., 2022) and NeuralSDE (Kidger et al., 2021) are notable as well. Additionally, diffusion model (Sohl-Dickstein et al., 2015; Ho et al., 2020)-based approaches like TS-Diffusion (Li, 2023) and the discrete/continuous stochastic process diffusions (DSPD/CSPD) (Biloš et al., 2023) have been explored.

A common aspect among many of these models is the introduction of differential equations to handle continuous time. For instance, ODE-RNN (Rubanova et al., 2019) or neural controlled differential equation (Kidger et al., 2020) is used to encode irregular observational data, and ordinary differential equations or SDEs are employed for the temporal evolution of latent variables. The numerical integration of such differential equations cannot be parallelized, which often leads to increased computation time.

Further limitations exist for specific models. For example, KoVAE can only generate time series with regular intervals, while NeuralSDE and TS-Diffusion are limited to unconditional generation tasks only. ACSSM is theoretically able to handle unconditional generation, but, as we show in this work, it is practically infeasible to train for such settings. The original CTFP imposes a strong Markovian constraint on the modeled process. DSPD, while flexible, often suffers from slower generation and higher memory cost due to the nature of diffusion sampling.

**State Space Models and Neural Network Extensions** SSMs provide a classical and probabilistic framework for modeling time series data (Anderson & Moore, 2005; Durbin & Koopman, 2012; Lin & Michailidis, 2024), and OUFlow can be interpreted as a member of this family. In data assimilation, basic Kalman filters (Kalman, 1960; Kalman & Bucy, 1961) and smoothers (Rauch et al., 1965), which assume linear Gaussian system and observation models, as well as extended versions like the Extended Kalman Filter (Anderson & Moore, 2005) and Unscented Kalman Filter (Wan & Van Der Merwe, 2000; Julier & Uhlmann, 2004) for handling more nonlinear systems, are commonly used. Incorporating neural networks to address complex, nonlinear, and non-Gaussian processes has become a natural extension (Lin & Michailidis, 2024). Examples include the Deep Kalman Filter (Krishnan et al., 2015) and Normalizing Kalman Filter (NKF) (de Bézenac et al., 2020) in discrete time, and LatentODE, LatentSDE, CRU (Schirmer et al., 2022), NCDSSM, and ACSSM in continuous time.

Recently, neural sequence models have incorporated state space model (SSM)-based architectures such as S4 (Gu et al., 2022) and its various extensions (Smith et al., 2023; Gu & Dao, 2024; Patro & Agneeswaran, 2025; Lv et al., 2025; Somvanshi et al., 2025). While these approaches showcase the effectiveness of SSMs in efficiently handling long-range dependencies in sequential data, they are typically deterministic and require additional techniques to construct probabilistic time-series generative models (Goel et al., 2022; Zhou et al., 2023).

While the mainstream approach centers on maximizing the evidence lower bound (ELBO) via amortized inference, NKF instead utilizes normalizing flows to allow direct likelihood evaluation. OUFlow can be regarded as a continuous-time extension of NKF, employing mixtures of Ornstein-Uhlenbeck processes for the latent variable dynamics. This design preserves model expressiveness while eliminating the need for computationally expensive time integration.

## 3 METHODOLOGY

In this section, we introduce the model architecture and the algorithms for time-series generation and training of OUFlow. The derivations of each result presented in this section are provided in Appendix A.

### 3.1 MODEL

Our proposed model, OUFlow, consists of three components: the mixture of OU processes, linear observation, and the normalizing flow as in Figure 1. Probability dis-

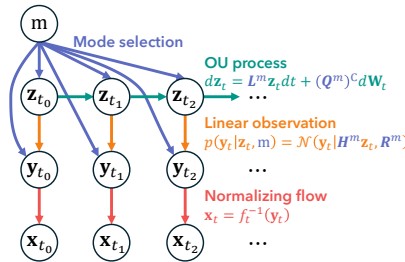

Figure 1: Graphical model of the proposed model, OUFlow.

tribution of the target variable $\mathbf{x}_t \in \mathbb{R}^d$ at time $t$ is generated by the normalizing flow from the latent variable $\boldsymbol{y}_t \in \mathbb{R}^d$, which is given by the linear observation of the other latent variable $\mathbf{z}_t \in \mathbb{R}^{d'}$ that follows the mixture of OU processes.

Given that $\boldsymbol{z}_t^m$ follows the OU process, which is a continuous-time stochastic process defined by the following stochastic differential equation:

$$d\mathbf{z}_t^m = \boldsymbol{L}^m \mathbf{z}_t^m dt + (\boldsymbol{Q}^m)^{\mathsf{C}} d\mathbf{W}_t^m \tag{1}$$

for all $m = 1, \cdots, M$, where $M$ is the hyperparameter indicating the number of mixed modes, $\boldsymbol{L}^m$ is the drift matrix, $\boldsymbol{Q}^m$ is the diffusion matrix, and each element of $\mathbf{W}_t^m$ is independent standard Wiener process. Throughout this paper, we use the superscript $\mathsf{C}$ to denote the Cholesky decomposition of a positive definite matrix. In OUFlow, the latent variable $\mathbf{z}_t$ follows the mixture of OU processes, which is defined by the following probability distribution:

$$p(\mathbf{z}_t) = \sum_{m=1}^{M} w_m p(\mathbf{z}_t|\mathrm{m}), \tag{2}$$

where $w_m := p(\mathrm{m})$ is the prior probability of the $m$-th mode and $p(\mathbf{z}_t|\mathrm{m}) = \delta(\mathbf{z}_t - \mathbf{z}_t^m)$. The objective of mixing OU processes is to extend the temporal evolution of the probability distribution of $\mathbf{z}_t$ beyond linear Gaussian models, which can significantly enhance the accuracy of data generation (see Appendix D.3).

One of the advantages of the OU process is that it has a closed-form solution for the transition probability, which is given by the following equation:

$$p(\mathbf{z}_{t'}|\mathbf{z}_t, \mathrm{m}) = \mathcal{N}\left(\mathbf{z}_{t'}|\xi_{t'-t}^m(\mathbf{z}_t), \boldsymbol{\Omega}_{t'-t}^m\right) \tag{3}$$

for all $t' \geq t$. Here, the mean $\boldsymbol{\xi}^m$ and the covariance matrix $\boldsymbol{\Omega}^m$ is given by

$$\boldsymbol{\xi}_{\Delta t}^m(\mathbf{z}_t) = e^{\boldsymbol{L}^m \Delta t} \mathbf{z}_t^m, \tag{4a}$$

$$\boldsymbol{\Omega}_{\Delta t}^m = \boldsymbol{P}^m \boldsymbol{S}_{\Delta t}^m \left(\boldsymbol{P}^m\right)^{\mathsf{T}}, \quad (S_{\Delta t}^m)_{ij} := \frac{e^{(\lambda_i^m + \lambda_j^m)\Delta t} - 1}{\lambda_i^m + \lambda_j^m} \left[\left(\boldsymbol{P}^m\right)^{-1} \boldsymbol{Q}^m \left(\boldsymbol{P}^m\right)^{-\mathsf{T}}\right]_{ij} \tag{4b}$$

with the eigenvalues $\lambda_i^m$ and the eigenvectors $\boldsymbol{P}^m$ of the drift matrix $\boldsymbol{L}^m$: $\boldsymbol{L}^m = \boldsymbol{P}^m \boldsymbol{\Lambda}^m \left(\boldsymbol{P}^m\right)^{-1}$ and $\boldsymbol{\Lambda}^m := \mathrm{diag}(\boldsymbol{\lambda}^m)$. Throughout this paper, we use the notation $\left(\boldsymbol{P}^m\right)^{-\mathsf{T}} := \left[\left(\boldsymbol{P}^m\right)^{\mathsf{T}}\right]^{-1}$.

Here, $\boldsymbol{\xi}_{\Delta t}^m(\mathbf{z}_t)$ and $\boldsymbol{\Omega}_{\Delta t}^m$ generally involve computationally expensive matrix exponentials. To address this, OUFlow improves computational efficiency by directly learning $\boldsymbol{\lambda}$ as parameters, rather than $\boldsymbol{L}$ itself. This allows for analytic expressions that bypass explicit matrix exponential computations (see Appendix B.1).

The initial distributions of $\mathbf{z}_t^m$ in OUFlow is chosen as the Gaussian distribution:

$$p(\mathbf{z}_0^m) = \mathcal{N}(\mathbf{z}_0^m|\boldsymbol{\mu}_0^m, \boldsymbol{\Sigma}_0^m). \tag{5}$$

As shown in the following subsections, this lets the posterior distribution of $\mathbf{z}_t$ be computed in a closed-form solution.

To obtain the target variable $\mathbf{x}_t$, the latent variable $\mathbf{z}_t$ is first transformed into the other latent variable $\mathbf{y}_t$ by the linear observation defined as

$$p(\mathbf{y}_t|\mathbf{z}_t, \mathrm{m}) = \mathcal{N}(\mathbf{y}_t|\boldsymbol{H}^m \mathbf{z}_t, \boldsymbol{R}^m) \tag{6}$$

with the observation matrix $\boldsymbol{H}^m \in \mathbb{R}^{d \times d'}$ and the observation noise covariance matrix $\boldsymbol{R}^m \in \mathbb{R}^{d \times d}$. Then, the latent variable $\mathbf{y}_t$ is transformed into the target variable $\mathbf{x}_t$ by the normalizing flow defined as

$$\mathbf{y}_t = \boldsymbol{f}_t(\mathbf{x}_t). \tag{7}$$

The normalizing flow $\boldsymbol{f}_t : \mathbb{R}^d \to \mathbb{R}^{d'}$ is a bijective function that maps $\mathbf{x}_t$ to $\mathbf{y}_t$ and can be conditional normalizing flow to be time-dependent.

To summarize, trainable parameters of OUFlow are the eigenvalues $\boldsymbol{\lambda}^m$, the diffusion matrix $\boldsymbol{Q}^m$, the prior probability $w_m$, the initial mean $\boldsymbol{\mu}_0^m$ and covariance $\boldsymbol{\Sigma}_0^m$, the observation matrix $\boldsymbol{H}^m$, the observation noise covariance $\boldsymbol{R}^m$, and the parameters in normalizing flow $\boldsymbol{f}_t$. Not all of these parameters take real values; for example, $\boldsymbol{Q}^m$ must be a real symmetric matrix. How to deal with these constraints is discussed in Appendix B.

## 3.2 FILTERING: LIKELIHOOD COMPUTATION

To train OUFlow, we need to compute the likelihood of the observed data $\boldsymbol{X} = \{\boldsymbol{X}_n\}_{n=1}^N$ at the chronologically-ordered observation times $\{t_n\}_{n=1}^N$. This can be done in a manner similar to the Kalman filter, which is a recursive algorithm used to estimate the state of a linear dynamical system from a series of noisy observations.

The resultant likelihood is given by

$$p(\boldsymbol{X}) = \sum_{m=1}^M w_m p(\boldsymbol{X}|\mathrm{m}), \quad p(\boldsymbol{X}|\mathrm{m}) = \prod_{n=1}^N \left| \det \frac{\partial \boldsymbol{f}_{t_n}(\boldsymbol{X}_n)}{\partial \boldsymbol{x}} \right| \mathcal{N}(\boldsymbol{f}_{t_n}(\boldsymbol{X}_n)|\boldsymbol{\nu}_n^m, \boldsymbol{\Upsilon}_n^m), \quad (8)$$

where $\boldsymbol{\nu}_n^m$ and $\boldsymbol{\Upsilon}_n^m$ are computed in closed-form from the observed data $\boldsymbol{X}$ (see Appendix A.2 for the explicit expressions and their derivations). It is important to note that, in general, the computation of the negative log-likelihood involves matrix operations in the target space of dimension $d$, resulting in a computational complexity of $O(d^3)$. However, by restricting the observation noise covariance matrix $\boldsymbol{R}^m$ to be diagonal and utilizing several matrix identities (Harville, 1998), this complexity can be reduced to $O(d)$.

## 3.3 SMOOTHING: POSTERIOR COMPUTATION

OUFlow can generate irregular time-series data from the learned prior distribution as well as forecast and impute from the posterior distribution given the observed data. The computation of the posterior distributuion is similar to the Rauch-Tung-Striebel smoother (Rauch et al., 1965), which is a backward recursion algorithm to compute the posterior distribution for linear dynamical systems.

Firstly, the posterior distribution of m is computed by the Bayes' theorem as

$$p(\mathrm{m}|\boldsymbol{X}) = \frac{w_m p(\boldsymbol{X}|\mathrm{m})}{\sum_{\mathrm{m}} w_m p(\boldsymbol{X}|\mathrm{m})} \quad (9)$$

with the likelihood $p(\boldsymbol{X}|\mathrm{m})$ given in Equation (8).

To generate posterior time-series data, the transition probability of $\mathbf{z}_t$ is required. For $t \geq t_N$, the transition probability is given by Equation (3). For time before the final observation $t_N$, the backward transition probability is given for $t$ and $t'$ in the same observation interval as

$$p(\mathbf{z}_t|\mathbf{z}_{t'}, \boldsymbol{X}, \mathrm{m}) = \mathcal{N}\left(\mathbf{z}_t|\bar{\boldsymbol{\xi}}_{t,t'}^m(\mathbf{z}_{t'}), \bar{\boldsymbol{\Omega}}_{t,t'}^m\right) \quad (t_{l(t)} \leq t \leq t' \leq t_{l(t)+1}), \quad (10)$$

where $\bar{\boldsymbol{\xi}}^m$ and $\bar{\boldsymbol{\Omega}}^m$ are available in closed form (see Appendix A.3), and $l(t)$ is the index of the last observation time before $t$, i.e., $t_{l(t)} = \max\{t_n \mid t_n \leq t\}$.

## 3.4 GENERATION

To generate time-series data from OUFlow, we first sample the mode m from the prior distribution $p(\mathrm{m})$ or the posterior distribution $p(\mathrm{m}|\boldsymbol{X})$ given in Equation (9). Then, the latent variable $\mathbf{z}_t$ is sampled, and after passing through a linear observation, a normalizing flow is applied to produce $\mathbf{x}_t$.

To sample the trajectory of the latent variable $\mathbf{z}_t$ at given time points $t \in \mathcal{T}_{\text{gen}}$, we first sample $\mathbf{z}_{t_N}$ from the posterior distribution $p(\mathbf{z}_{t_N}|\boldsymbol{X}, \mathrm{m})$ (or if there is no observation, $\mathbf{z}_0$ from the prior distribution $p(\mathbf{z}_0|\mathrm{m})$). Then, we sample $\{\mathbf{z}_t\}_{t \in \mathcal{T}_{\text{gen}}}$ for $t > t_N$ from the transition probability [Equation (3)] and $\{\mathbf{z}_t\}_{t \in \mathcal{T}_{\text{gen}}}$ for $t < t_N$ from the backward transition probability [Equation (10)] recursively. Although a straightforward implementation of this sampling procedure incurs an $O(N + K)$ computational complexity, where $K = |\mathcal{T}_{\text{gen}}|$, it can be reduced to $O(\log(N + K))$ via parallel computation using the scan algorithm (Blelloch, 1990) (see Appendix A.4 for details).

## 3.5 TRAINING

Training of OUFlow is performed by maximum likelihood estimation. Assume that we have $S$ scenarios of irregular time series $\{\boldsymbol{X}^s\}_{s=1}^S$ as the training data, where each scenario $\boldsymbol{X}^s$ consists of $N^s$ observations $\{\boldsymbol{X}_n^s\}_{n=1}^{N^s}$ at time points $\{t_n^s\}_{n=1}^{N^s}$. First, we determine the forecast horizon $T$ and randomly sample $s \in \{1, \cdots, S\}$ and $t \in [0, t_{N_s} - T]$ a total of $B$ times, where $B$ is the batch

size. For each sampled $s$ and $t$, create a mini-batch $\{\boldsymbol{X}_{\text{batch}}^b\}_{b=1}^B$ by extracting $\{\boldsymbol{X}_n^s\}_{n|t_n \in [t,t+T]}$, resulting in $B$ sets of irregular time-series data with a maximum time length of $T$. OUFlow is trained by minimizing the mean time-averaged negative log-likelihood

$$\mathcal{L} = -\frac{1}{B} \sum_{b=1}^B \frac{1}{N_{\text{batch}}^b} \log p(\boldsymbol{X}_{\text{batch}}^b), \tag{11}$$

where $N_{\text{batch}}^b$ is the number of observations in the $b$-th scenario in the mini-batch. This process is repeated until specified criteria are met, with resampling of $s$ and $t$.

However, in many cases, only a subset of modes is effectively learned, with the weights $w_m$ for all other modes approaching zero. This issue arises because, when there is a large scale disparity among the likelihood $p(\boldsymbol{X}|m)$ across modes, those with smaller likelihoods are numerically ignored within the loss function due to machine precision limitations (see Appendix B.4). To mitigate this, we found it beneficial to introduce auxiliary losses during the early stages of training to encourage mode diversity. Specifically, we add the negative entropy of the posterior mode distribution for each batch,

$$\mathcal{L}_{\text{balance}} = \frac{1}{B} \sum_{b=1}^B \sum_{m=1}^M p(\text{m}|\boldsymbol{X}_{\text{batch}}^b) \log p(\text{m}|\boldsymbol{X}_{\text{batch}}^b), \tag{12}$$

as well as the mean negative log-likelihood for each mode

$$\mathcal{L}_{\text{mode}} = -\frac{1}{B} \sum_{b=1}^B \frac{1}{M} \sum_{m=1}^M \log p(\boldsymbol{X}_{\text{batch}}^b|m) \tag{13}$$

as auxiliary objectives to promote mode utilization and enhance overall accuracy.

## 4 EXPERIMENTS

To assess OUFlow's performance, we compare its generation quality and computational efficiency against methods that handle generation, forecasting, and imputation for irregular time-series data. Specifically, we evaluate two variants of DSPD: DSPD-GP, which employs a Gaussian process with a radial basis function kernel as the noise process, and DSPD-OU, which employs an OU process. Although ACSSM was originally designed for forecasting and imputation, it is, in principle, capable of unconditional generation; accordingly, we adopt it as one of the baselines in our evaluation. For more detailed experimental settings, please refer to Appendix C, and for additional experimental results, see Appendix D.

### 4.1 EXPERIMENTAL SETUP

We evaluate OUFlow's performance across three tasks: generation, forecasting, and imputation. In generation, we assess its ability to create initial time data. For forecasting, we examine its accuracy in predicting future data points. In imputation, we evaluate its effectiveness in reconstructing missing data.

For all time series data except Lorenz63, we create training, validation, and test datasets by shuffling all time points, dividing them into three equal parts, and then reordering each part chronologically. Although some studies (Lai et al., 2018) use continuous period splits, which may align with certain real-world scenarios, they often lead to extrapolative tasks. Such tasks, like predicting winter weather with a model trained on summer data, can introduce confounding factors in model evaluation. Thus, we choose not to use this approach.

In the generation task, we evaluate the similarity between the model's prior distribution of initial conditions $p(\mathbf{x}_0)$ and the distribution of the test data. We employ the energy distance (Szekely & Rizzo, 2004; Baringhaus & Franz, 2004)

$$d_{\text{E}}(P,Q) := 2\mathbb{E}_{\mathbf{x}\sim P, \mathbf{y}\sim Q} \|\mathbf{x}-\mathbf{y}\| - \mathbb{E}_{\mathbf{x},\mathbf{x}'\sim P} \|\mathbf{x}-\mathbf{x}'\| - \mathbb{E}_{\mathbf{y},\mathbf{y}'\sim Q} \|\mathbf{y}-\mathbf{y}'\| \tag{14}$$

to represent the distance between the model's prior $P$ and the test data distribution $Q$. The expectation values for $P$ are approximated by the Monte Carlo method with 4096 samples, and those for $Q$

Table 2: Energy distances for initial generation of the proposed model and the baselines. Mean and standard deviation are reported over 64 trials.

| MODEL | LORENZ63 | EXCHANGE | WEATHER | SOLAR |
|---|---|---|---|---|
| GPR | 0.0275±0.0015 | 0.0927±0.0030 | 0.1387±0.0066 | 3.4742±0.0990 |
| LATENTSDE | 0.0595±0.0015 | 0.1394±0.0020 | 0.1421±0.0032 | 2.3689±0.0249 |
| DSPD-GP | 0.1729±0.0088 | 0.0829±0.0030 | 0.1610±0.0042 | 2.6167±0.0256 |
| DSPD-OU | 0.4336±0.0110 | 0.2152±0.0051 | 0.5848±0.0072 | 4.1105±0.0158 |
| ACSSM | 0.7624±0.0012 | 1.3433±0.0013 | 1.8680±0.0012 | 6.7506±0.0017 |
| OUFLOW | **0.0016±0.0007** | **0.0148±0.0019** | **0.0195±0.0036** | **0.4834±0.0580** |

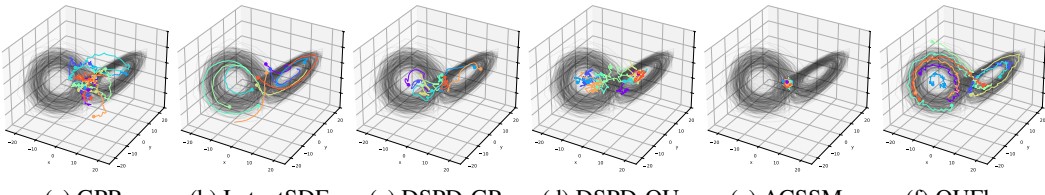

(a) GPR    (b) LatentSDE    (c) DSPD-GP    (d) DSPD-OU    (e) ACSSM    (f) OUFlow

Figure 2: Generated time-series data of Lorenz63 from the prior distribution. The black lines represent the test data trajectories. Each colored line represents a generated trajectory, whose initial state is represented as a point. 9 trajectories are generated from each initial state and the points generated at time intervals of 0.01 are continuously connected and plotted.

are approximated as the mean of the test data. We note that the quality of generation of time-series data is measured by this task together with the forecasting task.

In the forecasting task, we compare the posterior predictive distribution of the model at future time points, given several observed data points, with the test data. We sample 2048 slices with the forecast horizon $T$ randomly from the test data and then predict the last 2/3 of each slice with the first 1/3 as the observed data. We define time-averaged energy score (TAES) for model's predictive distribution $P$ as

$$\text{TAES}(P, \boldsymbol{X}) := \frac{1}{\sqrt{K}} \text{ES}_1(P, \boldsymbol{X}) \tag{15}$$

to evaluate the quality of the prediction, where $\text{ES}_\beta$ is the energy score, which is

$$\text{ES}_\beta(P, \boldsymbol{X}) = \mathbb{E}_{\mathbf{x} \sim P} \|\mathbf{x} - \boldsymbol{X}\|^\beta - \frac{1}{2} \mathbb{E}_{\mathbf{x}, \mathbf{x}' \sim P} \|\mathbf{x} - \mathbf{x}'\|^\beta \tag{16}$$

and strictly proper scoring rule for $\beta \in (0, 2)$ (Gneiting & Raftery, 2007), where $\boldsymbol{X}$ is the test data. We report the mean TAES and its standard deviation over valid slices out of 2048. The imputation task is similar to the forecasting task, but the observed data is first 1/6 and last 1/6 of each slice, and the middle 2/3 is predicted.

We use the following datasets for the experiments.

**Lorenz63** The Lorenz63 model is a three-variable system that evolves over time according to ordinary differential equations and is known for exhibiting chaotic behavior as a nonlinear system (Lorenz, 1963). In this study, we generated data through numerical simulations by adding noise to both the time evolution and the observations.

**Exchange** The dataset consists of daily exchange rate data from 1990 to 2016 for eight countries: Australia, the United Kingdom, Canada, Switzerland, China, Japan, New Zealand, and Singapore. The data was processed by (Lai et al., 2018).

**Weather** The dataset consists of weather information, represented by 21 real-valued variables (e.g., temperature), observed every 10 minutes at the Beutenberg Campus from January 1, 2023, to December 31, 2023[1].

**Solar** The dataset consists of power generation records from 137 PV plants in Alabama, with data collected every 10 minutes during the year 2006[2]. The data was processed by (Lai et al., 2018).

---

[1] https://www.bgc-jena.mpg.de/wetter/
[2] https://www.nrel.gov/grid/solar-power-data.html

Table 3: Mean TAES for forecasting of the proposed model and the baselines. Mean and standard deviation are reported over valid slices out of 2048.

| MODEL | LORENZ63 | EXCHANGE | WEATHER | SOLAR |
|---|---|---|---|---|
| GPR | 0.9475±0.3772 | 0.6634±0.3360 | 1.7413±0.9554 | 7.4672±1.9451 |
| LATENTSDE | 0.6398±0.4762 | 0.5202±0.2836 | 1.9656±1.4393 | 7.0281±6.7135 |
| DSPD-GP | 0.6142±0.2855 | 0.3793±0.2053 | 1.9152±1.2686 | 7.4792±4.7073 |
| DSPD-OU | 0.9905±0.4462 | 0.3097±0.1782 | 1.7777±1.2956 | 8.0489±5.2075 |
| ACSSM | 0.2897±0.1872 | 0.3341±0.1729 | 1.5426±1.0770 | **2.9982±2.7887** |
| OUFLOW | **0.1783±0.1464** | **0.2222±0.1447** | **1.3721±0.9943** | 4.1373±4.5752 |

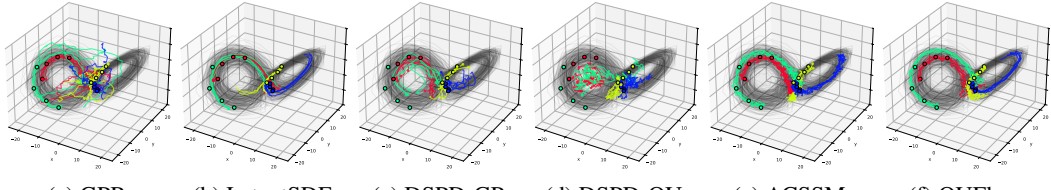

(a) GPR      (b) LatentSDE      (c) DSPD-GP      (d) DSPD-OU      (e) ACSSM      (f) OUFlow

Figure 3: Generated time-series data of Lorenz63 in the forecasting task. The black lines represent the test data trajectories. Each colored points represent the observed data, and the colored lines represent the corresponding generated trajectories. 4 patterns of observation points are shown, and 4 trajectories are generated from each observed data point.

### 4.2 RESULTS & DISCUSSION

#### 4.2.1 GENERATION

The energy distances for initial generation with the proposed model and the baselines are presented in Table 2. The proposed model OUFlow outperforms the baselines across all datasets. Generated time-series samples of Lorenz63 from the prior distribution are shown in Figure 2. It should be noted that, whereas the energy distance is used to evaluate the distributions of initial conditions, Figure 2 illustrates the entire generated trajectories.

Except for LatentSDE and OUFlow, none of the models sufficiently capture the nonlinear and non-Gaussian nature of the Lorenz63 trajectories. In particular, ACSSM fails to generate diverse paths in the unconditional generation setting, with most trajectories collapsing to the mean. Furthermore, as indicated by the energy distance, LatentSDE does not sufficiently replicate the prior distribution, resulting in biases in the generated trajectories. LatentSDE is a model based on VAE, and although the issue can be potentially mitigated by increasing the weight of the KL divergence in the loss function, as done in $\beta$-VAE (Higgins et al., 2017), this adjustment leads to a trade-off with reconstruction accuracy, generally reducing the quality of forecasting and imputation. OUFlow uniformly generates Lorenz63 trajectories and, being trained via maximum likelihood estimation, does not encounter the difficulty of balancing the loss function that is present in VAE-based models.

#### 4.2.2 FORECASTING & IMPUTATION

The mean TAES for forecasting and imputation using the proposed model and the baselines are presented in Table 3 and Table 4, respectively. OUFlow outperforms the baselines on the Lorenz63 and Exchange datasets, and achieves the best or competitive results on the remaining datasets. In summary, while ACSSM occasionally performs well on conditional generation tasks for high-dimensional data, OUFlow consistently achieves the best results across all other tasks.

Generated time-series samples of Lorenz63 for the forecasting and imputation tasks are shown in Figure 3 and Figure 4, respectively. Although LatentSDE appears to produce visually plausible trajectories, its energy score is worse than that of OUFlow, in part because LatentSDE underestimates the variability of the trajectories. In contrast, both ACSSM and OUFlow accurately capture not only the mean but also the fluctuations of the trajectories, resulting in superior energy scores.

#### 4.2.3 SCALING OF GENERATION TIME WITH GENERATED SEQUENCE LENGTH

Table 4: Same table as Table 3, but for the imputation task.

| MODEL | LORENZ63 | EXCHANGE | WEATHER | SOLAR |
|---|---|---|---|---|
| GPR | 0.6509±0.3180 | 0.2134±0.1278 | **1.1053±0.9551** | 5.8980±1.5030 |
| LATENTSDE | 0.6550±0.5093 | 0.3804±0.1707 | 1.6928±1.5971 | 5.7323±5.3026 |
| DSPD-GP | 0.2802±0.1292 | 0.3483±0.2034 | 1.7363±1.1893 | 8.3559±5.5370 |
| DSPD-OU | 0.8076±0.3934 | 0.2354±0.1324 | 1.5244±1.1801 | 7.5293±4.8169 |
| ACSSM | 0.2341±0.0997 | 0.2810±0.1343 | 1.2133±1.1478 | **1.7575±1.4351** |
| OUFLOW | **0.1005±0.0231** | **0.1491±0.0897** | 1.1238±1.3190 | 3.2993±3.3743 |

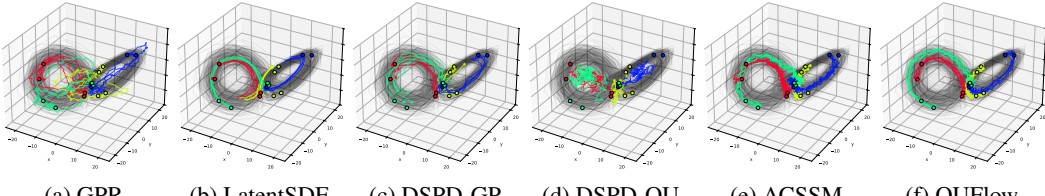

| (a) GPR | (b) LatentSDE | (c) DSPD-GP | (d) DSPD-OU | (e) ACSSM | (f) OUFlow |
|---|---|---|---|---|---|

Figure 4: Same figure as Figure 3, but for the imputation task.

The average wall-clock time[3] for generation of the proposed model and the baselines are shown in Figure 5. GPR and DSPD exhibit computational complexity that scales cubically with the number of generated points, $K$. Consequently, as $K$ increases, the generation time rises sharply; for DSPD, memory exhaustion occurred at $K = 10000$. LatentSDE, on the other hand, is limited by the time integration required from the start to the end of the generated trajectory. While its generation time is ideally independent of $K$ under perfect parallelization, in practice, parallelization efficiency decreases as $K$ increases, resulting in longer computation times. In contrast, neither ACSSM nor OUFlow requires time integration, and under ideal parallelization, their generation time scales logarithmically with $K$. In practice, as with LatentSDE, the efficiency of parallelization declines for large $K$, causing the actual scaling to deviate from perfect logarithmic growth. Nonetheless, both ACSSM and OUFlow consistently achieve relatively fast generation for any number of generated points. Considering that ACSSM is not suited to generation tasks, OUFlow exhibits the best scaling with respect to the number of generated time steps among models that support all three tasks.

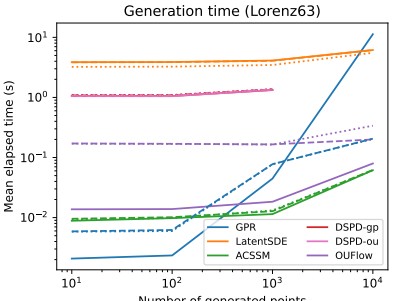

Figure 5: Average wall-clock time for generation of the proposed model and the baselines. The solid lines correspond to the generation from prior distribution, dashed lines correspond to the forecasting, and dotted lines correspond to the imputation. Setups are the same as Figure 2, Figure 3, and Figure 4, respectively. 16 trajectories are generated in parallel for each task and the wall-clock time is measured by the average of 64 trials.

## 5 FURTHER APPLICATIONS

Beyond conventional use cases for time-series generative models, OUFlow's analytical tractability opens the door to a wide range of applications. In this section, we discuss several potential directions that leverage the distinctive properties of OUFlow. Concrete experimental results on Lorenz63 are provided in Appendix E.

**Likelihood Evaluation**  As shown in Equation (8), OUFlow admits explicit likelihood evaluation for irregularly sampled time series, enabling applications such as anomaly detection and out-of-distribution detection. In particular, one can flag sequences with low likelihood as anomalies; such approaches are commonly referred to as density-based methods (Yang et al., 2024). Empirically, an OUFlow model trained on the Lorenz63 dataset assigns markedly lower likelihoods to artificially

---

[3]The measurements are conducted on a machine equipped with an Intel Core i7-12700 CPU and an NVIDIA GeForce RTX 2080 Ti GPU. The code is implemented using PyTorch (Paszke et al., 2019).

corrupted sequences (see Figure 7). That said, density-based methods can suffer from unstable density estimation, especially in high-dimensional settings, and may require corrective techniques such as likelihood ratios (Ren et al., 2019).

**Clustering**    Because OUFlow adopts a mixture model in the latent space, we expect its modes to align with distinct clusters in the data. Indeed, in our experiments on Lorenz63, the trained OUFlow appears to learn modes that correspond to different subsets of trajectories (see Figure 8). Moreover, given an irregularly sampled sequence, the posterior mode distribution $p(\mathrm{m}|\boldsymbol{X})$, computable via Equation (9), enables inference of the sequence's cluster assignment.

**Denoising**    In OUFlow, uncertainty decomposes into two components: (i) latent-process uncertainty arising from a finite $\boldsymbol{Q}$ in the OU dynamics, and (ii) observation uncertainty induced by a finite $\boldsymbol{R}$ in the linear observation model. Consequently, one can selectively suppress these noise sources during generation. When observation uncertainty in the linear measurement model is removed, i.e., when $\boldsymbol{R} = 0$, the conditional distribution $p(\mathbf{x}_{t'}|\mathbf{x}_t)$ converges to $\delta(\mathbf{x}_{t'} - \mathbf{x}_t)$ in the limit $t' \to t$; equivalently, the instantaneous variance at coincident times vanishes. The finite instantaneous variance eliminated in this setting corresponds, under appropriate assumptions, to observation noise and can therefore be suppressed independently of system noise (see the denoising example on Lorenz63 in Figure 9). It should be noted that these assumptions do not generally hold, and in such cases separating system noise from observation noise is challenging (Johansson et al., 1999; Kong et al., 2023).

## 6 CONCLUSION

In this paper, we proposed OUFlow, a novel general-purpose time series generative model. By combining two analytically tractable components, an Ornstein–Uhlenbeck (OU) process and a normalizing flow, OUFlow provides a unified model that consistently handles generation, forecasting, and imputation for irregular time series, while also opening avenues for applications such as anomaly detection, clustering, and denoising. Moreover, by leveraging scan-style parallelization, it achieves generation times that scale logarithmically with the length of the generated sequence. Experiments on both synthetic and real-world datasets demonstrate that, among models capable of all three tasks, OUFlow delivers state-of-the-art generation quality.

**Limitations & Future Work**    While the approach of combining a mixture distribution of OU processes with normalizing flows has empirically demonstrated some expressivity in this paper, its capacity to serve as a universal approximator remains unestablished. Moreover, because OUFlow involves operations such as matrix inversion for the latent variables, the computational complexity increases cubically with the dimensionality of these latent variables. Consequently, phenomena requiring inherently high-dimensional latent variables may lead to longer generation times. In addition, scaling with respect to the number of observations is linear, which can lead to substantial computational burden for large observation sets. To overcome these drawbacks, we will continue to explore latent variable temporal evolution models and decoders that offer higher expressivity and greater computational efficiency for posterior distribution calculations.

### THE USE OF LARGE LANGUAGE MODELS

We used an LLM for proofreading the manuscript and for portions of the related work survey. We also employed an LLM for ideation, including the design of the loss functions, and to help code the implementation of our ideas.

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

# A   DERIVATIONS

In this section, we derive the equations used in the main text. Before proceeding, we introduce several well-known formulas that are instrumental in these derivations.

We frequently utilize the following identity for the product of two normal distributions:

$$\mathcal{N}(\mathbf{x}|\boldsymbol{A}\mathbf{y} + \boldsymbol{b}, \boldsymbol{V})\mathcal{N}(\mathbf{y}|\boldsymbol{\mu}, \boldsymbol{\Sigma}) = \mathcal{N}(\mathbf{x}|\boldsymbol{A}\boldsymbol{\mu} + \boldsymbol{b}, \boldsymbol{A}\boldsymbol{\Sigma}\boldsymbol{A}^T + \boldsymbol{V})$$
$$\times \mathcal{N}(\mathbf{y}|\mu + \boldsymbol{K}\left(\mathbf{x} - \boldsymbol{A}\boldsymbol{\mu} - \boldsymbol{b}\right), (\boldsymbol{I} - \boldsymbol{K}\boldsymbol{A})\boldsymbol{\Sigma}), \quad (17)$$

where

$$\boldsymbol{K} := \boldsymbol{\Sigma}\boldsymbol{A}^T(\boldsymbol{A}\boldsymbol{\Sigma}\boldsymbol{A}^T + \boldsymbol{V})^{-1}, \quad (18)$$

and $\boldsymbol{A} \in \mathbb{R}^{d_1 \times d_2}$, $\boldsymbol{b} \in \mathbb{R}^{d_1}$, $\boldsymbol{V} \in \mathbb{R}^{d_1 \times d_1}$, $\boldsymbol{\mu} \in \mathbb{R}^{d_2}$, $\boldsymbol{\Sigma} \in \mathbb{R}^{d_2 \times d_2}$, $\mathbf{x} \in \mathbb{R}^{d_1}$, and $\mathbf{y} \in \mathbb{R}^{d_2}$ are arbitrary vectors and matrices. The left-hand side of Equation (17) facilitates marginalization with respect to $\mathbf{x}$, while the right-hand side is structured to easily allow marginalization with respect to $\mathbf{y}$.

We also employ Woodbury's identity:

$$(\boldsymbol{A} + \boldsymbol{U}\boldsymbol{C}\boldsymbol{V})^{-1} = \boldsymbol{A}^{-1} - \boldsymbol{A}^{-1}\boldsymbol{U}(\boldsymbol{C}^{-1} + \boldsymbol{V}\boldsymbol{A}^{-1}\boldsymbol{U})^{-1}\boldsymbol{V}\boldsymbol{A}^{-1} \quad (19)$$

and the matrix determinant lemma:

$$|\boldsymbol{A} + \boldsymbol{U}\boldsymbol{C}\boldsymbol{V}| = |\boldsymbol{A}||\boldsymbol{C}|\left|\boldsymbol{C}^{-1} + \boldsymbol{V}\boldsymbol{A}^{-1}\boldsymbol{U}\right| \quad (20)$$

that apply to arbitrary matrices $\boldsymbol{A} \in \mathbb{R}^{d_1 \times d_1}$, $\boldsymbol{U} \in \mathbb{R}^{d_1 \times d_2}$, $\boldsymbol{C} \in \mathbb{R}^{d_2 \times d_2}$, and $\boldsymbol{V} \in \mathbb{R}^{d_2 \times d_1}$ (Harville, 1998). These identities are employed to reduce the dimensions of the necessary matrix operations, thereby decreasing the computational load.

## A.1   TRANSITION PROBABILITY [EQUATIONS (3) AND (4)]

By defining $\zeta_t^m := e^{-\boldsymbol{L}^m}\mathbf{z}_t^m$, we can rewrite the SDE in Equation (1) as

$$d\zeta_t^m = e^{-\boldsymbol{L}^m t}(\boldsymbol{Q}^m)^{\mathsf{C}}d\mathbf{W}_t^m. \quad (21)$$

This equation can be integrated to yield

$$\zeta_{t'}^m = \zeta_t^m + \int_{s=t}^{t'} e^{-\boldsymbol{L}^m s}(\boldsymbol{Q}^m)^{\mathsf{C}}d\mathbf{W}_s^m \quad (22)$$

and transformed back to the original space as

$$\mathbf{z}_{t'}^m = e^{\boldsymbol{L}\Delta t}\mathbf{z}_t^m + \int_{s=0}^{\Delta t} e^{\boldsymbol{L}^m(\Delta t - s)}(\boldsymbol{Q}^m)^{\mathsf{C}}d\mathbf{W}_s^m \quad (23)$$

with $\Delta t := t' - t$. The second term of the left-hand side $\mathbf{i}_{\Delta t} := \int_{s=0}^{\Delta t} e^{\boldsymbol{L}^m(\Delta t - s)}(\boldsymbol{Q}^m)^{\mathsf{C}} d\mathbf{W}_s^m$ is a Gaussian random variable with mean $\mathbf{0}$ since it is a linear combination of Gaussian random variables with mean $\mathbf{0}$. Therefore, the transition probabiity $p(\mathbf{z}_{t'}^m | \mathbf{z}_t^m, \mathbf{m})$ is a Gaussian distribution with mean $e^{\boldsymbol{L}^m \Delta t} \mathbf{z}_t^m$ and covariance matrix

$$
\begin{aligned}
\boldsymbol{\Omega}_{\Delta t}^m &= \mathbb{E}\left[\mathbf{i}_{\Delta t} \mathbf{i}_{\Delta t}^{\mathsf{T}}\right] \\
&= \int_{s=0}^{\Delta t} \int_{s'=0}^{\Delta t} e^{\boldsymbol{L}^m(\Delta t - s)}(\boldsymbol{Q}^m)^{\mathsf{C}} \mathbb{E}[d\mathbf{W}_s^m d\mathbf{W}_{s'}^{m\mathsf{T}}](\boldsymbol{Q}^m)^{\mathsf{CT}} e^{(\boldsymbol{L}^m)^{\mathsf{T}}(\Delta t - s')} \\
&= \int_{s=0}^{\Delta t} \int_{s'=0}^{\Delta t} e^{\boldsymbol{L}^m(\Delta t - s)}(\boldsymbol{Q}^m)^{\mathsf{C}} \delta(s - s') ds ds' \boldsymbol{I}_{d'}(\boldsymbol{Q}^m)^{\mathsf{CT}} e^{(\boldsymbol{L}^m)^{\mathsf{T}}(\Delta t - s')} \\
&= \int_0^{\Delta t} ds\, e^{\boldsymbol{L}^m(\Delta t - s)} \boldsymbol{Q}^m e^{(\boldsymbol{L}^m)^{\mathsf{T}}(\Delta t - s)}.
\end{aligned}
\tag{24}
$$

From this result, we can derive the following identity:

$$
\begin{aligned}
\boldsymbol{L}^m \boldsymbol{\Omega}_{\Delta t}^m + \boldsymbol{\Omega}_{\Delta t}^m (\boldsymbol{L}^m)^{\mathsf{T}} &= \int_0^{\Delta t} ds\, \big[ \boldsymbol{L}^m e^{\boldsymbol{L}^m(\Delta t - s)} \boldsymbol{Q}^m e^{(\boldsymbol{L}^m)^{\mathsf{T}}(\Delta t - s)} \\
&\qquad\qquad + e^{\boldsymbol{L}^m(\Delta t - s)} \boldsymbol{Q}^m e^{(\boldsymbol{L}^m)^{\mathsf{T}}(\Delta t - s)} (\boldsymbol{L}^m)^{\mathsf{T}} \big] \\
&= -\int_0^{\Delta t} ds\, \frac{d}{ds}\left[ e^{\boldsymbol{L}^m(\Delta t - s)} \boldsymbol{Q}^m e^{(\boldsymbol{L}^m)^{\mathsf{T}}(\Delta t - s)} \right] \\
&= e^{\boldsymbol{L}^m \Delta t} \boldsymbol{Q}^m e^{(\boldsymbol{L}^m)^{\mathsf{T}} \Delta t} - \boldsymbol{Q}^m.
\end{aligned}
\tag{25}
$$

This equation is cast into the form of the Lyapunov equation:

$$
\boldsymbol{\Lambda}^m \boldsymbol{S}_{\Delta t}^m + \boldsymbol{S}_{\Delta t}^m \boldsymbol{\Lambda}^m = e^{\boldsymbol{\Lambda}^m \Delta t} (\boldsymbol{P}^m)^{-1} \boldsymbol{Q}^m (\boldsymbol{P}^m)^{-\mathsf{T}} e^{\boldsymbol{\Lambda}^m \Delta t} - (\boldsymbol{P}^m)^{-1} \boldsymbol{Q}^m (\boldsymbol{P}^m)^{-\mathsf{T}}
\tag{26}
$$

through the introduction of $\boldsymbol{S}_{\Delta t}^m := (\boldsymbol{P}^m)^{-1} \boldsymbol{\Omega}_{\Delta t}^m (\boldsymbol{P}^m)^{-\mathsf{T}}$. Dividing by $\lambda_i + \lambda_j$ from the $(i, j)$-element of this equation yields Equation (4).

## A.2 LIKELIHOOD [EQUATION (8)]

Here, we prove that the likelihood $p(\boldsymbol{X})$ is given by Equation (8) with explicit expressions for $\boldsymbol{\nu}_n^m$ and $\boldsymbol{\Upsilon}_n^m$. They are shown by recursively applying the following recursion relation to 1, which corresponds to $p(\boldsymbol{X}_{1:n} | \mathbf{m})$ for $n = 0$:

$$
p(\boldsymbol{X}_{1:n+1} | \mathbf{m}) = \left| \det \frac{\partial \boldsymbol{f}_{t_{n+1}}(\boldsymbol{X}_{n+1})}{\partial \boldsymbol{x}} \right| \mathcal{N}(\boldsymbol{f}_{t_{n+1}}(\boldsymbol{X}_{n+1}) | \boldsymbol{\nu}_{n+1}^m, \boldsymbol{\Upsilon}_{n+1}^m) p(\boldsymbol{X}_{1:n} | \mathbf{m}),
\tag{27}
$$

where $\boldsymbol{\nu}_n^m$ and $\boldsymbol{\Upsilon}_n^m$ are defined as

$$
\boldsymbol{\nu}_n^m = \boldsymbol{H}^m \hat{\boldsymbol{\mu}}_{t_n | n-1}^m,
\tag{28a}
$$

$$
\boldsymbol{\Upsilon}_n^m = \boldsymbol{H}^m \hat{\boldsymbol{\Sigma}}_{t_n | n-1}^m (\boldsymbol{H}^m)^{\mathsf{T}} + \boldsymbol{R}^m,
\tag{28b}
$$

$\hat{\boldsymbol{\mu}}_{t|n}^m$ and $\hat{\boldsymbol{\Sigma}}_{t|n}^m$ are the filtered mean and covariance of the latent variable $\mathbf{z}_t$ given the observed data $\boldsymbol{X}_{1:n}$ with $t \geq t_n$, respectively:

$$
p(\mathbf{z}_t | \boldsymbol{X}_{1:n}, \mathbf{m}) = \mathcal{N}(\mathbf{z}_t | \hat{\boldsymbol{\mu}}_{t|n}^m, \hat{\boldsymbol{\Sigma}}_{t|n}^m).
\tag{29}
$$

Also, $\hat{\boldsymbol{\mu}}_{t|n}^m$ and $\hat{\boldsymbol{\Sigma}}_{t|n}^m$ are computed from the filtered mean $\hat{\boldsymbol{\mu}}_n^m := \hat{\boldsymbol{\mu}}_{t_n|n}^m$ and covariance $\hat{\boldsymbol{\Sigma}}_n^m := \hat{\boldsymbol{\Sigma}}_{t_n|n}^m$ at the last observation time $t_n$ as

$$
\hat{\boldsymbol{\mu}}_{t|n}^m = e^{\boldsymbol{L}^m(t - t_n)} \hat{\boldsymbol{\mu}}_n^m,
\tag{30a}
$$

$$
\hat{\boldsymbol{\Sigma}}_{t|n}^m = e^{\boldsymbol{L}^m(t - t_n)} \hat{\boldsymbol{\Sigma}}_n^m e^{(\boldsymbol{L}^m)^{\mathsf{T}}(t - t_n)} + \boldsymbol{\Omega}_{t-t_n}^m.
\tag{30b}
$$

$\hat{\boldsymbol{\mu}}_n^m$ and $\hat{\boldsymbol{\Sigma}}_n^m$ are computed recursively from the initial mean $\hat{\boldsymbol{\mu}}_0^m = \boldsymbol{\mu}_0^m$ and covariance $\hat{\boldsymbol{\Sigma}}_0^m = \boldsymbol{\Sigma}_0^m$ by the following equations:

$$\hat{\boldsymbol{\mu}}_{n+1}^m = \hat{\boldsymbol{\mu}}_{t_{n+1}|n}^m + \tilde{\boldsymbol{K}}_{n+1}^m \left[ \tilde{\boldsymbol{f}}_{n+1} - \boldsymbol{\Pi}^m \hat{\boldsymbol{\mu}}_{t_{n+1}|n}^m \right], \tag{31a}$$

$$\hat{\boldsymbol{\Sigma}}_{n+1}^m = \left[ \boldsymbol{I} - \tilde{\boldsymbol{K}}_{n+1}^m \boldsymbol{\Pi}^m \right] \hat{\boldsymbol{\Sigma}}_{t_{n+1}|n}^m. \tag{31b}$$

where $\tilde{\boldsymbol{f}}$, $\tilde{\boldsymbol{K}}$ and $\boldsymbol{\Pi}^m$ are defined as

$$\tilde{\boldsymbol{f}}_n := (\boldsymbol{H}^m)^\mathsf{T} (\boldsymbol{R}^m)^{-1} \boldsymbol{f}_{t_n}(\boldsymbol{X}_n), \tag{32a}$$

$$\tilde{\boldsymbol{K}}_n^m := \hat{\boldsymbol{\Sigma}}_{t_n|n-1}^m \left\{ \boldsymbol{I} - \boldsymbol{\Pi}^m \left[ \left( \hat{\boldsymbol{\Sigma}}_{t_n|n-1}^m \right)^{-1} + \boldsymbol{\Pi}^m \right]^{-1} \right\}, \tag{32b}$$

$$\boldsymbol{\Pi}^m := (\boldsymbol{H}^m)^\mathsf{T} (\boldsymbol{R}^m)^{-1} \boldsymbol{H}^m. \tag{32c}$$

To derive these expressions, we use mathematical induction; assuming that the filtered probability $p(\mathbf{z}_{t_n}|\boldsymbol{X}_{1:n}, \mathrm{m})$ is Gaussian with mean $\hat{\boldsymbol{\mu}}_n^m$ and covariance $\hat{\boldsymbol{\Sigma}}_n^m$ for $n \geq 0$, we show that $p(\mathbf{z}_{t_{n+1}}|\boldsymbol{X}_{1:n+1}, \mathrm{m})$ is also Gaussian with mean and covariance given by Equations (31) and (32), while deriving Equations (29) and (30) and the recursion relation for the likelihood [Equations (27) and (28)].

First, for $n = 0$, the filtered probability is the prior distribution $p(\mathbf{z}_0|\mathrm{m}) = \mathcal{N}(\mathbf{z}_0|\boldsymbol{\mu}_0^m, \boldsymbol{\Sigma}_0^m)$, which is a Gaussian distribution. If $p(\mathbf{z}_{t_n}|\boldsymbol{X}_{1:n}, \mathrm{m}) = \mathcal{N}(\mathbf{z}_{t_n}|\hat{\boldsymbol{\mu}}_n^m, \hat{\boldsymbol{\Sigma}}_n^m)$ for $n \geq 0$, Equations (29) and (30) can be derived as

$$\begin{aligned} p(\mathbf{z}_t|\boldsymbol{X}_{1:n}, \mathrm{m}) &= \int d\mathbf{z}_{t_n} p(\mathbf{z}_t|\mathbf{z}_{t_n}, \mathrm{m}) p(\mathbf{z}_{t_n}|\boldsymbol{X}_{1:n}, \mathrm{m}) \\ &= \int d\mathbf{z}_{t_n} \mathcal{N}(\mathbf{z}_t|\xi_{t-t_n}^m(\mathbf{z}_{t_n}), \boldsymbol{\Omega}_{t-t_n}^m) \mathcal{N}(\mathbf{z}_{t_n}|\hat{\boldsymbol{\mu}}_n^m, \hat{\boldsymbol{\Sigma}}_n^m) \\ &= \mathcal{N}\left( \mathbf{z}_t | e^{\boldsymbol{L}^m(t-t_n)} \hat{\boldsymbol{\mu}}_n^m, e^{\boldsymbol{L}^m(t-t_n)} \hat{\boldsymbol{\Sigma}}_n^m e^{(\boldsymbol{L}^m)^\mathsf{T}(t-t_n)} + \boldsymbol{\Omega}_{t-t_n}^m \right). \end{aligned} \tag{33}$$

Here, we use Equation (17) to marginalize with respect to $\mathbf{z}_{t_n}$ in the second line. Then, from the graphical model given in Figure 1, the joint distribution of $\boldsymbol{X}_{1:n+1}$ and $\mathbf{z}_{t_{n+1}}$ can be expressed as

$$\begin{aligned} &p(\boldsymbol{X}_{1:n+1}, \mathbf{z}_{t_{n+1}}|\mathrm{m}) \\ &= p(\boldsymbol{X}_{n+1}|\mathbf{z}_{t_{n+1}}, \mathrm{m}) p(\mathbf{z}_{t_{n+1}}|\boldsymbol{X}_{1:n}, \mathrm{m}) p(\boldsymbol{X}_{1:n}|\mathrm{m}) \\ &= \left| \det \frac{\partial \boldsymbol{f}_{t_{n+1}}(\boldsymbol{X}_{n+1})}{\partial \boldsymbol{x}} \right| \mathcal{N}(f(\boldsymbol{X}_{n+1})|\boldsymbol{H}^m \mathbf{z}_{t_{n+1}}, \boldsymbol{R}^m) \mathcal{N}(\mathbf{z}_{t_{n+1}}|\hat{\boldsymbol{\mu}}_{t_{n+1}|n}^m, \hat{\boldsymbol{\Sigma}}_{t_{n+1}|n}^m) p(\boldsymbol{X}_{1:n}|\mathrm{m}) \\ &= \mathcal{N}\left( \mathbf{z}_{t_{n+1}} \left| \hat{\boldsymbol{\mu}}_{t_{n+1}|n}^m + \boldsymbol{K}_{n+1}^m \left[ \boldsymbol{f}_{t_{n+1}}(\boldsymbol{X}_{n+1}) - \boldsymbol{H}^m \hat{\boldsymbol{\mu}}_{t_{n+1}|n}^m \right], (\boldsymbol{I}_{d'} - \boldsymbol{K}_{n+1}^m \boldsymbol{H}^m) \hat{\boldsymbol{\Sigma}}_{t_{n+1}|n}^m \right. \right) \\ &\quad \times \left| \det \frac{\partial \boldsymbol{f}_{t_{n+1}}(\boldsymbol{X}_{n+1})}{\partial \boldsymbol{x}} \right| \mathcal{N}\left( f(\boldsymbol{X}_{n+1})|\boldsymbol{H}^m \hat{\boldsymbol{\mu}}_{t_{n+1}|n}^m, \boldsymbol{H}^m \hat{\boldsymbol{\Sigma}}_{t_{n+1}|n}^m (\boldsymbol{H}^m)^\mathsf{T} + \boldsymbol{R}^m \right) p(\boldsymbol{X}_{1:n}|\mathrm{m}), \end{aligned} \tag{34}$$

again utilizing Equation (17). On the other hand, this can also be decomposed as

$$p(\boldsymbol{X}_{1:n+1}, \mathbf{z}_{t_{n+1}}|\mathrm{m}) = p(\mathbf{z}_{t_{n+1}}|\boldsymbol{X}_{1:n+1}, \mathrm{m}) p(\boldsymbol{X}_{1:n+1}|\mathrm{m}). \tag{35}$$

By comparing these two equations, we obtain

$$\begin{aligned} &p(\mathbf{z}_{t_{n+1}}|\boldsymbol{X}_{1:n+1}, \mathrm{m}) \\ &= \mathcal{N}\left( \mathbf{z}_{t_{n+1}} \left| \hat{\boldsymbol{\mu}}_{t_{n+1}|n}^m + \boldsymbol{K}_{n+1}^m \left[ \boldsymbol{f}_{t_{n+1}}(\boldsymbol{X}_{n+1}) - \boldsymbol{H}^m \hat{\boldsymbol{\mu}}_{t_{n+1}|n}^m \right], (\boldsymbol{I}_{d'} - \boldsymbol{K}_{n+1}^m \boldsymbol{H}^m) \hat{\boldsymbol{\Sigma}}_{t_{n+1}|n}^m \right. \right), \end{aligned} \tag{36}$$

$$\begin{aligned} &p(\boldsymbol{X}_{1:n+1}|\mathrm{m}) \\ &= \left| \det \frac{\partial \boldsymbol{f}_{t_{n+1}}(\boldsymbol{X}_{n+1})}{\partial \boldsymbol{x}} \right| \mathcal{N}\left( f(\boldsymbol{X}_{n+1})|\boldsymbol{H}^m \hat{\boldsymbol{\mu}}_{t_{n+1}|n}^m, \boldsymbol{H}^m \hat{\boldsymbol{\Sigma}}_{t_{n+1}|n}^m (\boldsymbol{H}^m)^\mathsf{T} + \boldsymbol{R}^m \right) p(\boldsymbol{X}_{1:n}|\mathrm{m}), \end{aligned} \tag{37}$$

which yields recursion formula:

$$\hat{\boldsymbol{\mu}}_n^m = \hat{\boldsymbol{\mu}}_{t_n|n-1}^m + \boldsymbol{K}_n^m \left[ \boldsymbol{f}_{t_n}(\boldsymbol{X}_n) - \boldsymbol{H}^m \hat{\boldsymbol{\mu}}_{t_n|n-1}^m \right], \tag{38a}$$

$$\hat{\boldsymbol{\Sigma}}_n^m = (\boldsymbol{I}_{d'} - \boldsymbol{K}_n^m \boldsymbol{H}^m) \hat{\boldsymbol{\Sigma}}_{t_n|n-1}^m, \tag{38b}$$

with

$$\boldsymbol{K}_n^m := \hat{\boldsymbol{\Sigma}}_{t_n|n-1}^m (\boldsymbol{H}^m)^\mathsf{T} \left( \boldsymbol{H}^m \hat{\boldsymbol{\Sigma}}_{t_n|n-1}^m (\boldsymbol{H}^m)^\mathsf{T} + \boldsymbol{R}^m \right)^{-1} \tag{39}$$

and Equations (27) and (28).

The computation of $\hat{\boldsymbol{\mu}}_n^m$ and $\hat{\boldsymbol{\Sigma}}_n^m$ in Equation (38) requires calculating the Kalman gain $\boldsymbol{K}_n^m$ as in Equation (39), which incurs $O(d^3)$ computational cost with respect to the target space dimension $d$. By applying Woodbury's identity [Equation (19)], we can reformulate this as in Equations (31) and (32), thereby avoiding this expensive computation, and all of Equations (27) to (32) are derived.

From Equation (8), the negative log-likelihood is written as

$$- \log p(\boldsymbol{X}|\mathrm{m}) = \sum_{n=1}^N \left[ \frac{1}{2} \left\{ d \log(2\pi) + \log |\boldsymbol{\Upsilon}_n^m| + (\boldsymbol{f}_{t_n}(\boldsymbol{X}_n) - \boldsymbol{\nu}_n^m)^\mathsf{T} (\boldsymbol{\Upsilon}_n^m)^{-1} (\boldsymbol{f}_{t_n}(\boldsymbol{X}_n) - \boldsymbol{\nu}_n^m) \right\} \right.$$

$$\left. - \log \left| \det \frac{\partial \boldsymbol{f}_{t_n}(\boldsymbol{X}_n)}{\partial \boldsymbol{x}} \right| \right]. \tag{40}$$

This formulation also involves matrix operations with $O(d^3)$ computational complexity, resulting in poor computational efficiency. Therefore, by utilizing Equations (19) and (20), we derive an alternative representation that avoids these computational costs.

Poorly scaling parts of Equation (40) are rewritten as

$$|\boldsymbol{\Upsilon}_n^m| = \left| \boldsymbol{H}^m \hat{\boldsymbol{\Sigma}}_{t_n|n-1}^m (\boldsymbol{H}^m)^\mathsf{T} + \boldsymbol{R}^m \right|$$

$$= |\boldsymbol{R}^m| \left| \hat{\boldsymbol{\Sigma}}_{t_n|n-1}^m \right| \left| \left( \hat{\boldsymbol{\Sigma}}_{t_n|n-1}^m \right)^{-1} + \boldsymbol{\Pi}^m \right| \tag{41}$$

and

$$(\boldsymbol{f}_{t_n}(\boldsymbol{X}_n) - \boldsymbol{\nu}_n^m)^\mathsf{T} (\boldsymbol{\Upsilon}_n^m)^{-1} (\boldsymbol{f}_{t_n}(\boldsymbol{X}_n) - \boldsymbol{\nu}_n^m)$$

$$= \left( \boldsymbol{f}_{t_n}(\boldsymbol{X}_n) - \boldsymbol{H}^m \hat{\boldsymbol{\mu}}_{t_n|n-1}^m \right)^\mathsf{T} \left( \boldsymbol{H}^m \hat{\boldsymbol{\Sigma}}_{t_n|n-1}^m (\boldsymbol{H}^m)^\mathsf{T} + \boldsymbol{R}^m \right)^{-1} \left( \boldsymbol{f}_{t_n}(\boldsymbol{X}_n) - \boldsymbol{H}^m \hat{\boldsymbol{\mu}}_{t_n|n-1}^m \right)$$

$$= \left( \boldsymbol{f}_{t_n}(\boldsymbol{X}_n) - \boldsymbol{H}^m \hat{\boldsymbol{\mu}}_{t_n|n-1}^m \right)^\mathsf{T}$$

$$\times \left[ (\boldsymbol{R}^m)^{-1} - (\boldsymbol{R}^m)^{-1} \boldsymbol{H}^m \left\{ \left( \hat{\boldsymbol{\Sigma}}_{t_n|n-1}^m \right)^{-1} + \boldsymbol{\Pi}^m \right\}^{-1} (\boldsymbol{H}^m)^\mathsf{T} (\boldsymbol{R}^m)^{-1} \right]$$

$$\times \left( \boldsymbol{f}_{t_n}(\boldsymbol{X}_n) - \boldsymbol{H}^m \hat{\boldsymbol{\mu}}_{t_n|n-1}^m \right)$$

$$= \boldsymbol{f}_n^\mathsf{T}(\boldsymbol{X}_n) (\boldsymbol{R}^m)^{-1} \boldsymbol{f}_{t_n}(\boldsymbol{X}_n) - 2 \left( \hat{\boldsymbol{\mu}}_{t_n|n-1}^m \right)^\mathsf{T} \tilde{\boldsymbol{f}}_n + \hat{\boldsymbol{\mu}}_{t_n|n-1}^m \boldsymbol{\Pi}^m \hat{\boldsymbol{\mu}}_{t_n|n-1}^m$$

$$- \left( \tilde{\boldsymbol{f}}_n - \boldsymbol{\Pi}^m \hat{\boldsymbol{\mu}}_{t_n|n-1}^m \right)^\mathsf{T} \left\{ \left( \hat{\boldsymbol{\Sigma}}_{t_n|n-1}^m \right)^{-1} + \boldsymbol{\Pi}^m \right\}^{-1} \left( \tilde{\boldsymbol{f}}_n - \boldsymbol{\Pi}^m \hat{\boldsymbol{\mu}}_{t_n|n-1}^m \right). \tag{42}$$

Here, the definition of $\tilde{\boldsymbol{f}}$ and $\boldsymbol{\Pi}$ are given in Equation (32). These expressions does not require matrix operations with $O(d^3)$ computational time if $\boldsymbol{R}^m$ is diagonal, which is the case in this study. Additionally, by using $\tilde{\boldsymbol{K}}$ defined in Equation (32), we can similarly derive Equation (31) as a more computationally efficient expression for $\hat{\boldsymbol{\mu}}_n^m$ and $\hat{\boldsymbol{\Sigma}}_n^m$.

### A.3 POSTERIOR TRANSITION PROBABILITY [EQUATION (10)]

Posterior transition probability $p(\mathbf{z}_t|\mathbf{z}_{t'}, \boldsymbol{X}, \mathrm{m})$ for $t < t'$ is given as Equation (10) with

$$\bar{\boldsymbol{\xi}}_{t,t'}^m(\mathbf{z}_{t'}) = \hat{\boldsymbol{\mu}}_{t|l(t)}^m + \boldsymbol{F}_{t,t'}^m \left( \mathbf{z}_{t'} - \hat{\boldsymbol{\mu}}_{t'|l(t)}^m \right), \tag{43a}$$

$$\bar{\boldsymbol{\Omega}}_{t,t'}^m = \left( \boldsymbol{I} - \boldsymbol{F}_{t,t'}^m e^{\boldsymbol{L}^m(t'-t)} \right) \hat{\boldsymbol{\Sigma}}_{t|l(t)}^m, \tag{43b}$$

$$\boldsymbol{F}_{t,t'}^m := \hat{\boldsymbol{\Sigma}}_{t|l(t)}^m e^{(\boldsymbol{L}^m)^\mathsf{T}(t'-t_{l(t)})} \left( \hat{\boldsymbol{\Sigma}}_{t'|l(t)}^m \right)^{-1}. \tag{43c}$$

This is shown as follows.

From the graphical model given in Figure 1, the posterior transition probability $p(\mathbf{z}_t|\mathbf{z}_{t'}, \boldsymbol{X}, \mathrm{m})$ is calculated as

$$\begin{aligned} p(\mathbf{z}_t|\mathbf{z}_{t'}, \boldsymbol{X}, \mathrm{m}) &= p(\mathbf{z}_t|\mathbf{z}_{t'}, \boldsymbol{X}_{1:l(t)}, \mathrm{m}) \\ &\propto p(\mathbf{z}_{t'}|\mathbf{z}_t, \mathrm{m}) p(\mathbf{z}_t|\boldsymbol{X}_{1:l(t)}, \mathrm{m}) \\ &= \mathcal{N} \left( \mathbf{z}_{t'}|\xi_{t'-t}^m(\mathbf{z}_t), \boldsymbol{\Omega}_{t'-t}^m \right) \mathcal{N} \left( \mathbf{z}_t|\hat{\boldsymbol{\mu}}_{t|l(t)}^m, \hat{\boldsymbol{\Sigma}}_{t|l(t)}^m \right). \end{aligned} \tag{44}$$

By using Equation (17) to factor out all the $\mathbf{z}_t$ dependencies as normal distribution, we obtain Equations (10) and (43).

### A.4 GENERATION

Here, we describe an efficient method for fast sampling of latent variable sequences $\{\mathbf{z}_t\}_{t \in \mathcal{T}_{\mathrm{gen}}}$ from the posterior distribution given observed data $\{\boldsymbol{X}_n\}_{n=1}^N$. We define $\{\tau_k\}_{k=1}^{N+K}$ as the clonologically ordered time points of the union of generated time points $\mathcal{T}_{\mathrm{gen}}$ and observation time points $\{t_n\}_{n=1}^N$. We also let $\tilde{N}$ be the index of $\tau$ satisfying $\tau_{\tilde{N}} = t_N$. For $k \geq \tilde{N}$, the latent variable transformation

$$\tilde{\mathbf{z}}_{\tau_k}^m := e^{-L^m(\tau_k - t_N)} \mathbf{z}_{\tau_k}^m \tag{45}$$

yields the recursive formula

$$\tilde{\mathbf{z}}_{\tau_{k+1}}^m = \tilde{\mathbf{z}}_{\tau_k}^m + e^{-L^m(\tau_{k+1}-t_N)} \boldsymbol{\Omega}_{\tau_{k+1}-\tau_k}^{\mathsf{C}} \mathbf{r}_{k+1}^m \tag{46}$$

with the initial condition

$$\tilde{\mathbf{z}}_{\tau_{\tilde{N}}}^m = \hat{\boldsymbol{\mu}}_N^m + \left( \hat{\boldsymbol{\Sigma}}_N^m \right)^{\mathsf{C}} \mathbf{r}_{\tilde{N}}^m, \tag{47}$$

where $\mathbf{r}_k \sim \mathcal{N}(\mathbf{0}, \boldsymbol{I})$ for $k = 0, \cdots, N + K$. Since Equation (46) is the form of cumulative sum, it can be computed within $O(\log(N + K))$ time complexity with parallel computation by using the scan algorithm (Blelloch, 1990).

For $k < \tilde{N}$, the latent variable transformation

$$\tilde{\mathbf{z}}_{\tau_k}^m := \left( \boldsymbol{F}_{\tau_k, t_N}^m \right)^{-1} \mathbf{z}_{\tau_k}^m \tag{48}$$

also yields the recursive formula

$$\tilde{\mathbf{z}}_{\tau_{k-1}}^m = \tilde{\mathbf{z}}_{\tau_k}^m - \left( \boldsymbol{F}_{\tau_k, t_N}^m \right)^{-1} \hat{\boldsymbol{\mu}}_{\tau_k|m} + \left( \boldsymbol{F}_{\tau_{k-1}, t_N}^m \right)^{-1} \left[ \hat{\boldsymbol{\mu}}_{\tau_{k-1}|m} + \bar{\boldsymbol{\Omega}}_{\tau_{k-1}, \tau_k}^{\mathsf{C}} \mathbf{r}_{k-1} \right], \tag{49}$$

which can also be computed by the scan algorithm. Here, we define $\boldsymbol{F}_{t,t'}^m$ for any $t$ and $t'$ by

$$\boldsymbol{F}_{t,t_N}^m := \boldsymbol{F}_{t,t_{l(t)+1}}^m \boldsymbol{F}_{t_{l(t)+1}, t_{l(t)+2}}^m \cdots \boldsymbol{F}_{t_{l(t')-1}, t_{l(t')}}^m \boldsymbol{F}_{t_{l(t')}, t'}^m \tag{50}$$

as the extension of $\boldsymbol{F}_{t,t'}^m$ originally defined in Equation (43) for $t$ and $t'$ in the same observation interval. The bottleneck in computing $\boldsymbol{F}_{\tau_k,t'}^m$ for all $k < \tilde{N}$ lies in the calculation of $\boldsymbol{F}_{\tau_1,t'}^m$, which has a time complexity of $O(N)$.

## B   IMPLEMENTATION DETAILS

### B.1   TRANSITION PROBABILITY

The OU process given in Equation (1) has two parameters, $\boldsymbol{L}^m$ and $\boldsymbol{Q}_m$, for each mode $m$. Since $\boldsymbol{Q}^m$ must be a positive definite matrix, we treat the lower triangular elements of the cholesky decomposition $(\boldsymbol{Q}^m)^{\mathsf{C}}$ as trainable parameters; that is, $\boldsymbol{Q}^m = (\boldsymbol{Q}^m)^{\mathsf{C}} (\boldsymbol{Q}^m)^{\mathsf{CT}}$.

In the computation of transition probabilities and other calculations in OUFlow, the exponential matrix of $\boldsymbol{L}^m$ frequently appears, which can be easily computed using the diagonalization $\boldsymbol{L}^m = \boldsymbol{P}^m \boldsymbol{\Lambda}^m (\boldsymbol{P}^m)^{-1}$. The expression for $\boldsymbol{\Omega}^m$ in Equation (4) is also given using the diagonalization of $\boldsymbol{L}^m$. Therefore, instead of directly making $\boldsymbol{L}^m$ trainable parameters, we construct its eigenvalues $\boldsymbol{\lambda}^m$ as trainable parameters. The eigenvectors $\boldsymbol{P}^m$ define the basis of the latent space, and their degrees of freedom can be absorbed by the redefinitions of $\boldsymbol{Q}^m$ and $\boldsymbol{H}^m$. Thus, fixing $\boldsymbol{P}^m$ to a specific matrix does not result in a loss of generality.

For each component of $\boldsymbol{L}^m$ to be a real number, $\boldsymbol{\lambda}$ must appear as a complex conjugate pair. In this study, we limit the number of modes $M$ to be even, and the eigenvalues are paired as $\boldsymbol{\lambda}^m = \left\{ -\gamma_1^m + i\omega_1^m, -\gamma_1^m - i\omega_1^m, \ldots, -\gamma_{d'/2}^m + i\omega_{d'/2}^m, -\gamma_{d'/2}^m - i\omega_{d'/2}^m \right\}$. In this case, the corresponding eigenvectors must also form a complex conjugate pair, so we set

$$\boldsymbol{P}^m = \begin{bmatrix} \boldsymbol{J} & \boldsymbol{O} & \cdots & \boldsymbol{O} \\ \boldsymbol{O} & \boldsymbol{J} & \cdots & \boldsymbol{O} \\ \vdots & \vdots & \ddots & \vdots \\ \boldsymbol{O} & \boldsymbol{O} & \cdots & \boldsymbol{J} \end{bmatrix}, \tag{51}$$

where $\boldsymbol{O}$ is a $2 \times 2$ zero matrix and

$$\boldsymbol{J} := \frac{1}{2} \begin{bmatrix} 1 & 1 \\ -i & i \end{bmatrix}. \tag{52}$$

Under this choice of $\boldsymbol{P}^m$, the matrix $\boldsymbol{L}^m$ becomes a block-diagonal matrix $\boldsymbol{L}^m = \mathrm{diag}\left(\boldsymbol{L}_1^m, \ldots, \boldsymbol{L}_{d'/2}^m\right)$, where

$$\boldsymbol{L}_i^m = \begin{bmatrix} -\gamma_i^m & -\omega_i^m \\ \omega_i^m & -\gamma_i^m \end{bmatrix}, \tag{53}$$

which yields

$$e^{\boldsymbol{L}^m t} = \mathrm{diag}\left(e^{\boldsymbol{L}_1^m t}, \ldots, e^{\boldsymbol{L}_{d'/2}^m t}\right) \tag{54}$$

with

$$e^{\boldsymbol{L}_i^m t} = \begin{bmatrix} e^{-\gamma_i^m t} \cos(\omega_i^m t) & -e^{-\gamma_i^m t} \sin(\omega_i^m t) \\ e^{-\gamma_i^m t} \sin(\omega_i^m t) & e^{-\gamma_i^m t} \cos(\omega_i^m t) \end{bmatrix}. \tag{55}$$

After some algebra, we can derive the expression for $\boldsymbol{\Omega}_{\Delta t}^m$ as

$$\boldsymbol{\Omega}_{\Delta t}^m = \begin{bmatrix} \boldsymbol{\Omega}_{\Delta t;1,1}^m & \boldsymbol{\Omega}_{\Delta t;1,2}^m & \cdots & \boldsymbol{\Omega}_{\Delta t;1,d'/2}^m \\ \boldsymbol{\Omega}_{\Delta t;2,1}^m & \boldsymbol{\Omega}_{\Delta t;2,2}^m & \cdots & \boldsymbol{\Omega}_{\Delta t;2,d'/2}^m \\ \vdots & \vdots & \ddots & \vdots \\ \boldsymbol{\Omega}_{\Delta t;d'/2,1}^m & \boldsymbol{\Omega}_{\Delta t;d'/2,2}^m & \cdots & \boldsymbol{\Omega}_{\Delta t;d'/2,d'/2}^m \end{bmatrix} \tag{56}$$

where $2 \times 2$ matrix $\boldsymbol{\Omega}_{\Delta t;i,j}^m$ is given by

$$\boldsymbol{\Omega}_{\Delta t;i,j}^m = \frac{1}{2} \begin{bmatrix} \alpha_{\mathrm{r}\Delta t;ij}^m + \beta_{\mathrm{r}\Delta t;ij}^m & \alpha_{\mathrm{i}\Delta t;ij}^m - \beta_{\mathrm{i}\Delta t;ij}^m \\ \alpha_{\mathrm{i}\Delta t;ij}^m + \beta_{\mathrm{i}\Delta t;ij}^m & -\alpha_{\mathrm{r}\Delta t;ij}^m + \beta_{\mathrm{r}\Delta t;ij}^m \end{bmatrix}. \tag{57}$$

The terms $\alpha^m_{\text{r}\Delta t;ij}$, $\alpha^m_{\text{i}\Delta t;ij}$, $\beta^m_{\text{r}\Delta t;ij}$, and $\beta^m_{\text{i}\Delta t;ij}$ are defined as

$$
\alpha^m_{\text{r}\Delta t;ij} := \frac{1}{(\gamma^m_{+ij})^2 + (\omega^m_{+ij})^2}
$$
$$
\times \left[ \left( e^{-\gamma^m_{+ij}t} \cos \omega^m_{+ij}t - 1 \right) \left( -\gamma^m_{+ij}Q^m_{-ij} + \omega^m_{+ij}q^m_{+ij} \right) \right.
$$
$$
\left. + e^{-\gamma^m_{+ij}t} \sin \omega^m_{+ij}t \left( \gamma^m_{+ij}q^m_{+ij} + \omega^m_{+ij}Q^m_{-ij} \right) \right], \tag{58}
$$

$$
\alpha^m_{\text{i}\Delta t;ij} := \frac{1}{(\gamma^m_{+ij})^2 + (\omega^m_{+ij})^2}
$$
$$
\times \left[ - \left( e^{-\gamma^m_{+ij}t} \cos \omega^m_{+ij}t - 1 \right) \left( \gamma^m_{+ij}q^m_{+ij} + \omega^m_{+ij}Q^m_{-ij} \right) \right.
$$
$$
\left. + e^{-\gamma^m_{+ij}t} \sin \omega^m_{+ij}t \left( -\gamma^m_{+ij}Q^m_{-ij} + \omega^m_{+ij}q^m_{+ij} \right) \right], \tag{59}
$$

$$
\beta^m_{\text{r}\Delta t;ij} := \frac{1}{(\gamma^m_{+ij})^2 + (\omega^m_{-ij})^2}
$$
$$
\times \left[ \left( e^{-\gamma^m_{+ij}t} \cos \omega^m_{-ij}t - 1 \right) \left( -\gamma^m_{+ij}Q^m_{+ij} + \omega^m_{-ij}q^m_{-ij} \right) \right.
$$
$$
\left. + e^{-\gamma^m_{+ij}t} \sin \omega^m_{-ij}t \left( \gamma^m_{+ij}q^m_{-ij} + \omega^m_{-ij}Q^m_{+ij} \right) \right], \tag{60}
$$

$$
\beta^m_{\text{i}\Delta t;ij} := \frac{1}{(\gamma^m_{+ij})^2 + (\omega^m_{-ij})^2}
$$
$$
\times \left[ - \left( e^{-\gamma^m_{+ij}t} \cos \omega^m_{-ij}t - 1 \right) \left( \gamma^m_{+ij}q^m_{-ij} + \omega^m_{-ij}Q^m_{+ij} \right) \right.
$$
$$
\left. + e^{-\gamma^m_{+ij}t} \sin \omega^m_{-ij}t \left( -\gamma^m_{+ij}Q^m_{+ij} + \omega^m_{-ij}q^m_{-ij} \right) \right], \tag{61}
$$

where $\gamma^m_{\pm ij} := \gamma^m_i \pm \gamma^m_j$, $\omega^m_{\pm ij} := \omega^m_i \pm \omega^m_j$, $Q^m_{\pm ij} := Q^m_{2i-1,2j-1} \pm Q^m_{2i,2j}$, and $q^m_{\pm ij} := q^m_{2i,2j-1} \pm q^m_{2i-1,2j}$; they also give the expression of $\boldsymbol{S}^m_{\Delta t}$ in Equation (4) as

$$
\boldsymbol{S}^m_{\Delta t} = \begin{bmatrix} \boldsymbol{S}^m_{\Delta t;1,1} & \boldsymbol{S}^m_{\Delta t;1,2} & \cdots & \boldsymbol{S}^m_{\Delta t;1,d'/2} \\ \boldsymbol{S}^m_{\Delta t;2,1} & \boldsymbol{S}^m_{\Delta t;2,2} & \cdots & \boldsymbol{S}^m_{\Delta t;2,d'/2} \\ \vdots & \vdots & \ddots & \vdots \\ \boldsymbol{S}^m_{\Delta t;d'/2,1} & \boldsymbol{S}^m_{\Delta t;d'/2,2} & \cdots & \boldsymbol{S}^m_{\Delta t;d'/2,d'/2} \end{bmatrix} \tag{62}
$$

where $2 \times 2$ matrix $\boldsymbol{S}^m_{\Delta t;i,j}$ is given by

$$
S^m_{\Delta t;,i,j} = \frac{1}{2} \begin{bmatrix} \alpha^m_{\text{r}\Delta t;ij} + i\alpha^m_{\text{i}\Delta t;ij} & \beta^m_{\text{r}\Delta t;ij} + i\beta^m_{\text{i}\Delta t;ij} \\ \beta^m_{\text{r}\Delta t;ij} - i\beta^m_{\text{i}\Delta t;ij} & \alpha^m_{\text{r}\Delta t;ij} - i\alpha^m_{\text{i}\Delta t;ij} \end{bmatrix}. \tag{63}
$$

When some of the decay rates $\boldsymbol{\gamma}$ are negative, the latent variable $\mathbf{z}_t$ tends to grow over time, which we have observed can destabilize numerical computations of some covariance matrices. To address this, we use a trainable parameter $\boldsymbol{\theta} \in \mathbb{R}^{d'/2}$ and set $\gamma_i = \text{softplus}(\theta_i)$, thereby ensuring that $\gamma_i$ remains positive.

## B.2  LINEAR OBSERVATION

All elements of the linear transformation $\boldsymbol{H}^m$ in Equation (6) are trainable parameters. As mentioned in Appendix A.2, the observation noise $\boldsymbol{R}^m$ is assumed to be diagonal, and we treat their logarithm as trainable parameters:

$$
R_{ii} = \exp(\theta_i), \quad \boldsymbol{\theta}_i \in \mathbb{R}^{d'}. \tag{64}
$$

Sampling of $\mathbf{y}_t$ from Equation (6) is performed using the reparameterization trick (Kingma, 2013);

$$\mathbf{y}_t = \boldsymbol{H}^m \mathbf{z}_t + \exp\left(\frac{\boldsymbol{\theta}}{2}\right) \circ \epsilon, \quad \epsilon \sim \mathcal{N}(\mathbf{0}, \boldsymbol{I}_{d'}). \tag{65}$$

where $\circ$ denotes the element-wise product and $\exp$ is applied element-wise.

### B.3 NORMALIZING FLOW

The normalizing flow $\boldsymbol{f}_t$ used in OUFlow acts on the target variable, and since its dimension $d$ can become large, we employed RealNVP (Dinh et al., 2017), which allows for efficient computation of the Jacobian with respect to the increase in $d$. Additionally, to enable the flow to change with time $t$, we modified it to a conditional RealNVP by adding an extra dimension to each affine coupling layer to input the time, ensuring that it does not affect the bijectivity between $\mathbf{x}_t$ and $\mathbf{y}_t$.

### B.4 TRAINING

The direct computation of negative log-likelihood given in Equation (11) is numerically difficult because each $p(\boldsymbol{X}|\mathrm{m})$ is negative exponential of Equation (40), which can easily overflow. To avoid this, we use the following trick:

$$-\log p(\boldsymbol{X}) = -\log\left[\sum_{m=1}^{M} w_m p(\boldsymbol{X}|\mathrm{m})\right]$$

$$= c + \log\left[\sum_{m=1}^{M} w_m \exp\left(-\log p(\boldsymbol{X}|\mathrm{m}) - c\right)\right], \tag{66}$$

where $c$ is the dominant term defined as

$$c = \max_m \left[-\log p(\boldsymbol{X}|\mathrm{m})\right]. \tag{67}$$

By using this expression, each exponential term does not overflow because it is ensured to be less than or equal to 1.

### B.5 NUMERICAL ERROR

OUFlow involves many matrix operations including matrix multiplication, matrix inversion and Cholesky decomposition, which can be numerically unstable if the matrices are ill-conditioned. Therefore, using double precision is often crucial for improving generation accuracy.

Additionally, matrices that are theoretically guaranteed to be positive definite, such as covariance matrices, can sometimes become non-positive definite due to numerical errors, potentially preventing operations like Cholesky decomposition. To maintain the positive definiteness of such matrices, after certain matrix operations, we perform a check for positive definiteness. If the matrix is found not to be positive definite, we regularize it by adding $\epsilon \boldsymbol{I}_{d'}$, where $\epsilon$ is a small positive number.

## C EXPERIMENTAL DETAILS

### C.1 DATASETS

Table 5 summarizes the datasets used in this study. The "forecast horizon" represents the trained time span for which predictions can be made, while the "unit time" refers to the unit of time used to nondimensionalize the dataset. Each dataset is generated and preprocessed as follows.

Table 5: Datasets used in the experiments.

|  | # OF DIMS. | OBSERVATION INTERVAL | FORECAST HORIZON | UNIT TIME |
|---|---|---|---|---|
| LORENZ63 | 3 | 0.05 | 0.5 | 1 |
| EXCHANGE | 8 | 1 DAY | 30 DAYS | 30 DAYS |
| WEATHER | 22 | 10 MINUTES | 4 HOURS | 4 HOURS |
| SOLAR | 137 | 10 MINUTES | 4 HOURS | 4 HOURS |

### C.1.1   LORENZ63

To generate the dataset trajectory $\mathbf{x}_t$, we first numerically solve the following SDEs:

$$\frac{d\tilde{\mathbf{x}}_1}{dt} = \sigma(\tilde{\mathbf{x}}_2 - \tilde{\mathbf{x}}_1) + \xi_1, \tag{68a}$$

$$\frac{d\tilde{\mathbf{x}}_2}{dt} = \tilde{\mathbf{x}}_1(\rho - \tilde{\mathbf{x}}_3) - \tilde{\mathbf{x}}_2 + \xi_2, \tag{68b}$$

$$\frac{d\tilde{\mathbf{x}}_3}{dt} = \tilde{\mathbf{x}}_1\tilde{\mathbf{x}}_2 - \beta\tilde{\mathbf{x}}_3 + \xi_3, \tag{68c}$$

where $\tilde{\xi}$ is a vector of white noise with $\mathbb{E}\left[\xi_i(t)\xi_j(t')\right] = q\delta_{ij}\delta(t - t')$ and $\sigma = 10$, $\rho = 28$, $\beta = 8/3$. We also add observation noise $\eta$ to the trajectory $\tilde{\mathbf{x}}$:

$$\mathbf{x} = \tilde{\mathbf{x}} + \eta, \quad \eta \sim \mathcal{N}(\mathbf{0}, r^2\boldsymbol{I}_3). \tag{69}$$

We set the amplitudes of noises to $q = 1.0$ and $r = 0.5$.

The initial condition is fixed at $\tilde{\mathbf{x}}_0 = (1, 0, 0)^\mathsf{T}$, and simulations are run from $t = 0$ to $t = 30$ a total of 100 times. The trajectories are then clipped at $t < 10$ to prepare 100 trajectories within the range $t \in [10, 30]$. Out of these, 70 trajectories are used for training, 10 for validation, and 20 for testing. Data from each trajectory is recorded at intervals of 0.05 time units.

To stabilize learning by making the data approximate a standard normal distribution, a preprocessing step is applied. Specifically, $(0, 0, 24)^\mathsf{T}$ is subtracted from the data $\mathbf{x}$, and then each component is divided by 8.

### C.1.2   EXCHANGE

The exchange rate data consists entirely of positive real numbers, while the machine learning model used in this study is designed to handle values across the entire range of real numbers. Therefore, as a preprocessing step, the logarithm of the exchange rates was taken to transform their range to real numbers, and then each component is normalized to have a mean of 0 and a standard deviation of 1.

### C.1.3   WEATHER

The weather dataset consists of time series data with 21 features, such as temperature and humidity, each with different units and ranges. In this study, to evaluate the model's capability to handle such "messy" data, we applied only normalization to each feature as a preprocessing step.

### C.1.4   SOLAR

The Solar dataset consists of data on solar power generation, which is non-negative real numbers and includes a large number of zeros corresponding to times when no power is generated, such as during the night. Therefore, the following preprocessing steps are applied to each of the 137 components. For the zero values, we use the smallest non-zero value $x_{\min}$ to convert these to random variables within the range $(0, x_{\min})$ by sampling $x = x_{\min}\eta$ with $\eta \sim \text{Beta}(2, 4)$. Subsequently, all data are transformed to span the entire real number range using the function $\log(e^x - 1)$, and then normalization was applied.

## C.2 MODEL ARCHITECTURES AND TRAINING DETAILS

### C.2.1 OUFLOW

OUFlow's parameters are optimized using AdamW (Loshchilov & Hutter, 2019) with weight decay of $10^{-5}$. The initial learning rate for the normalizing flow parameters is selected via grid search from $\{0.0005, 0.001, 0.002, 0.005, 0.01, 0.02, 0.05\}$, while for the other parameters, the next higher value in the candidate set is used. The learning rates decay by 0.5 if the validation loss does not improve for 256 epochs. If the validation loss does not improve for 512 epochs, the training is stopped.

The latent dimension is set to $d' = 8$, the number of modes is set to $M = 16$, and the normalizing flow consists of 9 affine coupling layers. Each of these layers includes a 3-layer neural network with a hidden dimension of $4d$ and the ReLU activation function. As mentioned in Appendix B.5, the model was implemented using double precision to improve numerical stability.

During training, each mini-batch was constructed by randomly extracting 64 subsequences from the training data, each with a length corresponding to the forecast horizon, and then randomly dropping each time point with a probability of 0.5. Additionally, since the training data is provided at regular intervals, fixing the initial time would result in the model learning only specific time points. To address this, half of the 64 subsequences always start at time zero, while the other half have their initial time randomly shifted.

### C.2.2 GPR

GPR is implemented using GPyTorch (Gardner et al., 2018). We use MultitaskKernel with RBF kernel, and parameters are optimized by maximizing the likelihood. Adam (Kingma, 2014) with an initial learning rate of 0.1 is used as the optimizer and the learning rate is decayed by 0.5 if the validation loss does not improve for 16 epochs. If the validation loss does not improve for 32 epochs, the training is stopped. Batch size for training is set to 32.

### C.2.3 LATENTSDE

We use the implementation of LatentSDE included in torchsde library (Li et al., 2020). However, the original implementation does not support irregular time-series because the encoder was implemented as a GRU. To treat irregular time-series, we modified the encoder to be a GRU-D (Che et al., 2018), which is a GRU variant that can handle irregular time-series.

The optimizer used is Adam, and the initial learning rate is chosen from $\{0.0005, 0.001, 0.002, 0.005, 0.01, 0.02, 0.05\}$ by grid search. KL annealing is applied to the loss function, and the KL weight is increased from 0 to 1 linearly over the first `kl_anneal_iters` epochs. The learning rate is decayed by `lr_gamma` every epoch, and training is stopped after `num_iters` epochs. Hyperparameters `kl_anneal_iters`, `lr_gamma`, and `num_iters` are summarized in Table 6.

Table 6: Hyperparameters for LatentSDE.

| DATASET | NUM_ITERS | KL_ANNEAL_ITERS | LR_GAMMA | CONTEXT_SIZE | HIDDEN_SIZE |
|---|---|---|---|---|---|
| LORENZ63 | 5,000 | 1,000 | 0.997 | 24 | 48 |
| EXCHANGE | 10,000 | 3,000 | 0.999 | 48 | 96 |
| WEATHER | 10,000 | 3,000 | 0.999 | 128 | 256 |
| SOLAR | 20,000 | 6,000 | 0.9995 | 768 | 1536 |

The training data of Lorenz63 is regular time-series, while the other datasets are irregular and hence variable-length time series. The LatentSDE implementation does not support variable-length time series in one batch, so the batch size is set to 64 for Lorenz63 and 1 for the other datasets. The latent dimension is fixed to 8 and `context_size` and `hidden_size` are summarized in Table 6. Since OUFlow utilizes double precision, the numbers of model parameters in LatentSDE, DSPD, and ACSSM are configured to be about twice that of OUFlow to ensure a fair comparison. The comparison of the number of parameters is provided in Table 7.

Table 7: Number of model parameters.

| DATASET | MODEL | # OF PARAMS. |
|---------|-------|-------------:|
| LORENZ63 | GPR | 14 |
| | LATENTSDE | 17,975 |
| | DSPD | 17,699 |
| | ACSSM | 11,667 |
| | OUFLOW | 5,772 |
| EXCHANGE | GPR | 34 |
| | LATENTSDE | 65,528 |
| | DSPD | 57,352 |
| | ACSSM | 53,896 |
| | OUFLOW | 27,416 |
| WEATHER | GPR | 86 |
| | LATENTSDE | 433,987 |
| | DSPD | 349,749 |
| | ACSSM | 447,925 |
| | OUFLOW | 167,934 |
| SOLAR | GPR | 550 |
| | LATENTSDE | 15,129,059 |
| | DSPD | 15,014,537 |
| | ACSSM | 13,940,937 |
| | OUFLOW | 6,808,818 |

## C.2.4 DSPD

The original implementation of DSPD (Biloš et al., 2023) does not support arbitrary numbers of generated time points. Therefore, we re-implemented the noise estimator using a Transformer (Vaswani et al., 2017), enabling encoding of arbitrary observation points and decoding at any desired generation points. The number of diffusion steps during training was set to 1000, following Ho et al. (2020). For generation, we used 250 diffusion steps to reduce generation time and stabilize sample quality. As reported by Nichol & Dhariwal (2021), reducing the number of generation steps does not significantly degrade sample quality, and we observed that, especially for high-dimensional datasets, using fewer steps helps prevent divergence in the reverse diffusion process. The noise schedule follows Ho et al. (2020), employing a linear schedule from 1e-4 to 0.02. The hyperparameters for the Transformer are summarized in Table 8.

Table 8: Hyperparameters for the Transformer in DSPD.

| DATASET | HIDDEN_DIM | NUM_ENCODER_LAYERS | NUM_DECODER_LAYERS | DIM_FEEDFORWARD |
|---------|-----------:|-------------------:|-------------------:|----------------:|
| LORENZ63 | 16 | 3 | 3 | 32 |
| EXCHANGE | 24 | 4 | 4 | 64 |
| WEATHER | 48 | 6 | 6 | 144 |
| SOLAR | 256 | 8 | 8 | 1024 |

Training is performed using the AdamW optimizer with a weight decay of $10^{-5}$, and the batch size is set to 64. The initial learning rate is selected from $\{0.0001, 0.0002, 0.0005, 0.001, 0.002, 0.005, 0.01\}$ by grid search. The learning rate is reduced by a factor of 0.5 if the loss did not improve for 1280 epochs, and training is terminated if there was no improvement for 2560 epochs. For DSPD and ACSSM, which require training under various observation conditions including unconditional generation, mini-batches are constructed in the same manner as OUFlow but without random dropping. For each batch, an observation rate is sampled from the Beta distribution $\text{Beta}(1, 2)$, clipped at a minimum of 0.05, and observation points are randomly selected according to this rate.

### C.2.5 ACSSM

The original implementation of ACSSM (Park et al., 2025) contains several critical bugs that compromise consistency with the theoretical formulation. Therefore, we re-implement ACSSM based on the correct theoretical formulation; otherwise, we observe that, despite being theoretically capable, the generation accuracy significantly deteriorates when the number of generated points is changed from those used during training. We make the following changes to the `LinearSDE` class relative to the original implementation:

- In `parallel_compute` method, we correct a bug where a quantity that must be the time increment $\Delta t$ had been mistakenly implemented as the absolute time $t$, replacing $t$ with $\Delta t$ as required.

- In `parallel_compute` method, we correct an error where the time and batch dimensions are swapped when running the scan algorithm.

- In `parallel_compute` method, we remove the addition of `history` to `means`, as it violates associativity.

- In `get_matrix` method, we remove the `1e-6` regularization term, as it violates associativity.

- In the original implementation, $\alpha$—the embedding of each observation—is treated as the control policy over the interval immediately preceding that observation, which leaves the post-final-observation control policy undefined. We instead take the first observation to occur at $t = 0$ and interpret each $\alpha$ as the control policy over the interval immediately following its corresponding observation. To provide a control policy prior to the first observation, we introduce a dummy all-zero observation at time -1.

Training is performed using the same strategy as DSPD. The hyperparameters for ACSSM are summarized in Table 9.

Table 9: Hyperparameters for ACSSM.

| DATASET | STATE_DIM | N_LAYER | NUM_BASIS |
|---------|-----------|---------|-----------|
| LORENZ63 | 16 | 4 | 64 |
| EXCHANGE | 32 | 6 | 64 |
| WEATHER | 96 | 6 | 64 |
| SOLAR | 512 | 7 | 64 |

## D  ADDITIONAL EXPERIMENTAL RESULTS

### D.1  MEAN EUCLIDEAN DISTANCE

Table 10: MTAED measured in the generation task with the same setup as Table 2.

| MODEL | LORENZ63 | EXCHANGE | WEATHER | SOLAR |
|-------|----------|----------|---------|-------|
| GPR | 2.2296±0.0074 | 3.6896±0.0128 | 6.3635±0.0196 | 22.158±0.1811 |
| LATENTSDE | 2.1728±0.0059 | 3.7989±0.0101 | 5.8425±0.0134 | 13.145±0.0262 |
| DSPD-GP | 2.1684±0.0042 | 3.3912±0.0064 | 5.2316±0.0123 | 11.154±0.0188 |
| DSPD-OU | 1.9532±0.0027 | 3.1527±0.0042 | 4.6235±0.0048 | 10.706±0.0098 |
| ACSSM | **1.6507±0.0002** | **2.7108±0.0001** | **4.2190±0.0002** | **6.7506±0.0017** |
| OUFLOW | 2.3226±0.0071 | 3.5938±0.0085 | 6.2334±0.0417 | 17.161±0.2530 |

In the main text, the energy distance and energy score are used to evaluate the model's performance. While energy-based metrics are theoretically well-founded, their interpretation may not be immediately intuitive. To aid understanding, we further elaborate on these measures and additionally report the mean Euclidean distance as a complementary metric.

Table 11: MTAED measured in the forecasting task with the same setup as Table 3.

| MODEL | LORENZ63 | EXCHANGE | WEATHER | SOLAR |
|---|---|---|---|---|
| GPR | 1.8553±0.3870 | 1.3972±0.4580 | 3.7690±1.1146 | 20.922±3.1198 |
| LATENTSDE | 0.8334±0.5296 | 0.6361±0.3373 | 2.2374±1.4407 | 7.3718±6.7758 |
| DSPD-GP | 0.8856±0.3230 | 0.6895±0.2613 | 2.7078±1.3486 | 9.7729±4.7542 |
| DSPD-OU | 1.3788±0.4956 | 0.5331±0.1934 | 2.3839±1.3787 | 10.353±5.2079 |
| ACSSM | **0.4160±0.1874** | 0.5340±0.1730 | **1.8681±1.0771** | **3.8307±2.7894** |
| OUFLOW | 0.3652±0.1995 | **0.4372±0.2150** | 2.827±1.6218 | 7.2767±6.5798 |

Table 12: MTAED measured in the imputation task with the same setup as Table 4.

| MODEL | LORENZ63 | EXCHANGE | WEATHER | SOLAR |
|---|---|---|---|---|
| GPR | 1.3611±0.3185 | 0.4788±0.1722 | 2.4742±0.9736 | 17.307±3.3547 |
| LATENTSDE | 0.7853±0.5141 | 0.4541±0.1984 | 1.9222±1.5972 | 5.9777±5.3255 |
| DSPD-GP | 0.4866±0.1627 | 0.6629±0.2647 | 2.5186±1.2777 | 10.660±5.5424 |
| DSPD-OU | 1.2158±0.4642 | 0.4564±0.1511 | 2.1273±1.2630 | 9.7944±4.8652 |
| ACSSM | 0.3592±0.0999 | 0.4811±0.1344 | **1.5402±1.1481** | **2.5899±1.4375** |
| OUFLOW | **0.2086±0.0272** | **0.2941±0.1417** | 2.3389±1.6242 | 6.5900±5.5860 |

First, the energy distance is a metric for measuring the distance between distribution functions and satisfies the axioms of a distance measure. Consequently, the minimum value $d_{\mathrm{E}}(P, Q) = 0$ is satisfied if and only if the two distribution functions $P$ and $Q$ are identical. The energy distance can be evaluated when samples from both $P$ and $Q$ are available. In generation tasks, sampling from $Q$ can be performed by random sampling from the test data. However, in forecasting and imputation tasks, only one sample can be obtained from the test data for each observation. Therefore, considering that the third term of the energy distance given in Equation (14) is a constant independent of the model's generative distribution, the energy score, which evaluates the remaining two terms using the single test data $\boldsymbol{X}$, is used in this study, with its average across various conditions serving as the metric. For forecasting and imputation, since the length of the test data varies depending on the time slice used, the scores are normalized by dividing by $\sqrt{K}$ to ensure scale consistency, thereby obtaining the TAES.

If the constant term is ignored, the energy distance and energy score can be divided into two components: the "interaction term" and the "self-energy term". The interaction term defined as

$$\mathrm{MTAED} = \frac{1}{|S|} \sum_{s \in S} \frac{1}{\sqrt{K_s}} \mathbb{E}_{\mathbf{x} \sim P} \|\mathbf{x} - \boldsymbol{X}_s\| \tag{70}$$

corresponds to the first terms in Equation (14) and Equation (16), which essentially represents the mean time-averaged Euclidean distance (MTAED). Here, $S$ is the set of samples from the test data and $K_s$ is the number of generated timestamps in the sample $s$ ($K_s = 1$ for generation tasks). On the other hand, the self-energy term defined as

$$-\frac{1}{\sqrt{K}} \mathbb{E}_{\mathbf{x}, \mathbf{x}' \sim P} \|\mathbf{x} - \mathbf{x}'\| \tag{71}$$

corresponds to the second terms in Equation (14) and Equation (16), and it decreases as the variance of the generative distribution increases. Consequently, the energy distance and energy score naturally decrease when the average distance between the data generated by the model and the test data is small, while also penalizing the lack of diversity in the generated data.

To deepen the understanding of each model's tendencies and their effects on the energy distance and energy score, Tables 10 to 12 summarize the MTAED for the generation, forecasting, and imputation tasks, respectively. Although ACSSM consistently achieves the best MTAED across all datasets in the generation task, it ranks lowest in terms of energy distance. In practice, ACSSM tends to repeatedly generate samples near the mean (see Figure 2), failing to capture the true data distribution, which suggests that energy distance is a more appropriate metric. Similar observations hold for the forecasting and imputation tasks. For example, LatentSDE often achieves favorable MTAED scores across many datasets, but its TAES performance is suboptimal. As seen in the generated trajectories

for Lorenz63 (see Figures 3 and 4), LatentSDE tends to underestimate the variance, leading to this discrepancy.

It is worth noting that although the discriminative score and predictive score (Yoon et al., 2019) are used in some studies to evaluate the performance of time-series generative models, we do not use them in this paper because they are not sufficient to evaluate the quality of probability distribution of generated data. The discriminative score is a metric that evaluates the indiscriminability using a trained classifier that uses both real data and synthetic data generated by the model to be evaluated. Therefore, in extreme cases, even if a model that outputs only the training data is used, the score can be high, and it cannot measure the generalization performance or the diversity of generated scenarios. The predictive score, which evaluates the model performance using train-synthesis-test-real protocol, is also a metric that disregards the diversity of generated scenarios due to its evaluation metric being mean absolute error.

## D.2 GENERATION TIME

A comparison of generation times for datasets other than Lorenz63 is summarized in Figure 6. Similar trends to those observed in Lorenz63 can be confirmed across all datasets.

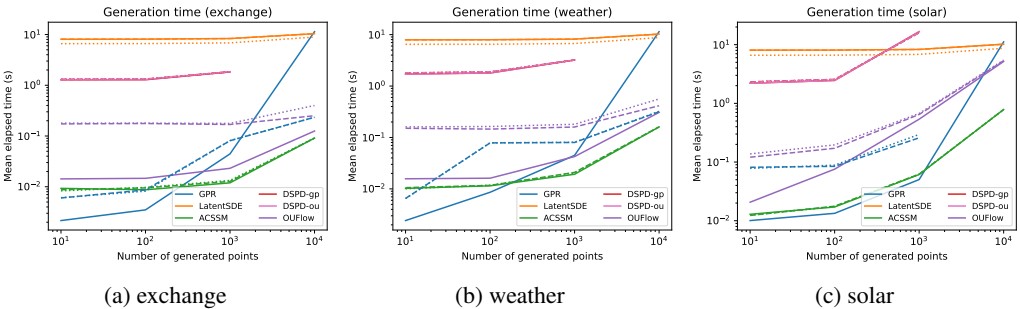

(a) exchange        (b) weather        (c) solar

Figure 6: Same figure as Figure 5 but for the Exchange, Weather, and Solar datasets.

## D.3 NUMBER OF MODES

In the architecture of OUFlow, an important factor for improving accuracy is the structure that mixes OU processes. Table 13 shows the mode number dependency of the energy distance and TAES for the Lorenz63 dataset. The case of $M = 1$ corresponds to using a single OU process, and it is evident that increasing the number of modes significantly improves the scores.

Table 13: $M$-dependence of energy distances between the true and generated distributions for generation tasks and TAES for forecasting and imputation tasks for Lorenz63. The experimental setups are the same as in Table 2, Table 3, and Table 4.

| $M$ | GENERATION | FORECASTING | IMPUTATION |
|---|---|---|---|
| 1 | 0.0188±0.0027 | 0.3576±0.2214 | 0.1456±0.0676 |
| 2 | 0.0061±0.0011 | 0.2658±0.2565 | 0.1163±0.0335 |
| 4 | 0.0053±0.0012 | 0.2709±0.2510 | 0.1083±0.0322 |
| 8 | 0.0015±0.0007 | 0.2240±0.2120 | 0.1050±0.0259 |
| 16 | 0.0016±0.0007 | 0.1783±0.1464 | 0.1005±0.0231 |
| 32 | 0.0009±0.0009 | 0.1636±0.1281 | 0.0981±0.0222 |
| 64 | 0.0007±0.0008 | 0.1644±0.1347 | 0.0980±0.0235 |

# E  FURTHER APPLICATIONS

## E.1  LIKELIHOOD EVALUATION

Figure 7 illustrates likelihood evaluation using OUFlow trained on the Lorenz63 dataset. We generated four sequences from Mode 2 (see Figure 8) at a temporal resolution of 0.01, and then injected two types of noise into the first sequence, yielding a total of six sequences for comparison. Specifically, we tested: (i) additive white noise $\xi(t)$ independently applied to all components with covariance $\mathbb{E}[\xi(t)\xi(t')] = 16\delta(t - t')$ (dashed line), and (ii) a point perturbation that adds 12 to $z(0.25)$ (dot-dashed line). As shown, the sequences with injected noise exhibit a pronounced drop in time-averaged log-likelihood relative to the clean sequences.

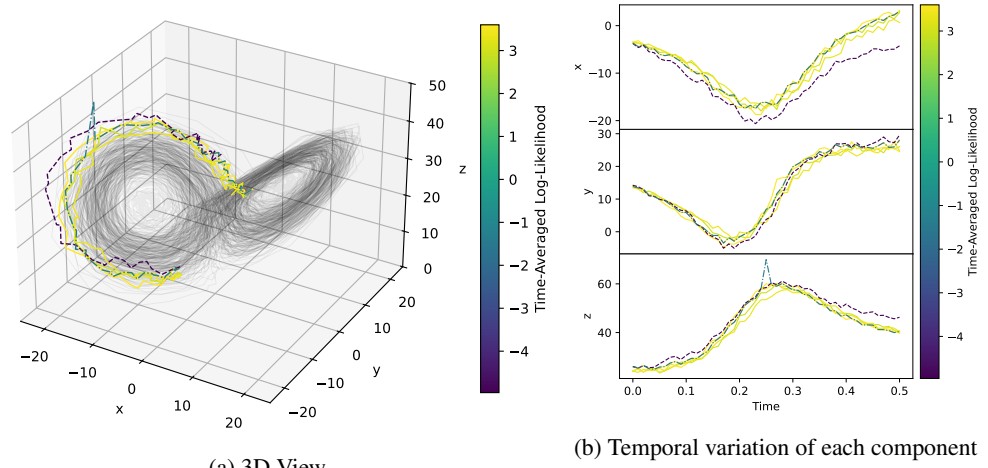

(a) 3D View

(b) Temporal variation of each component

Figure 7: Time-averaged log-likelihoods evaluated by OUFlow trained on Lorenz63. The solid lines represent four clean sequences generated from Mode 2 of OUFlow. The dashed lines represent the sequences with additive white noise with $a = 0.5$. The dot-dashed lines represent the sequences with a point perturbation adding 10 to $z(0.25)$.

## E.2  CLUSTERING

Figure 8 shows 16 samples generated from each mode of OUFlow trained on Lorenz63. Each mode predominantly captures a distinct dynamical regime of the Lorenz63 system.

## E.3  DENOISING

Figure 9 presents trajectories generated using an OUFlow model trained on the Lorenz63 dataset, where we selectively set $\boldsymbol{Q} = 0$ and/or $\boldsymbol{R} = 0$. Given the characteristics of the system noise $\xi$ and observation noise $\eta$ in Lorenz63 (see Appendix C.1.1), setting $\boldsymbol{Q} = 0$ is expected to suppress only the system noise, while setting $\boldsymbol{R} = 0$ removes only the observation noise. The figure juxtaposes trajectories produced by OUFlow with those generated under various noise conditions in Lorenz63, showing that the anticipated denoising behavior is largely achieved.

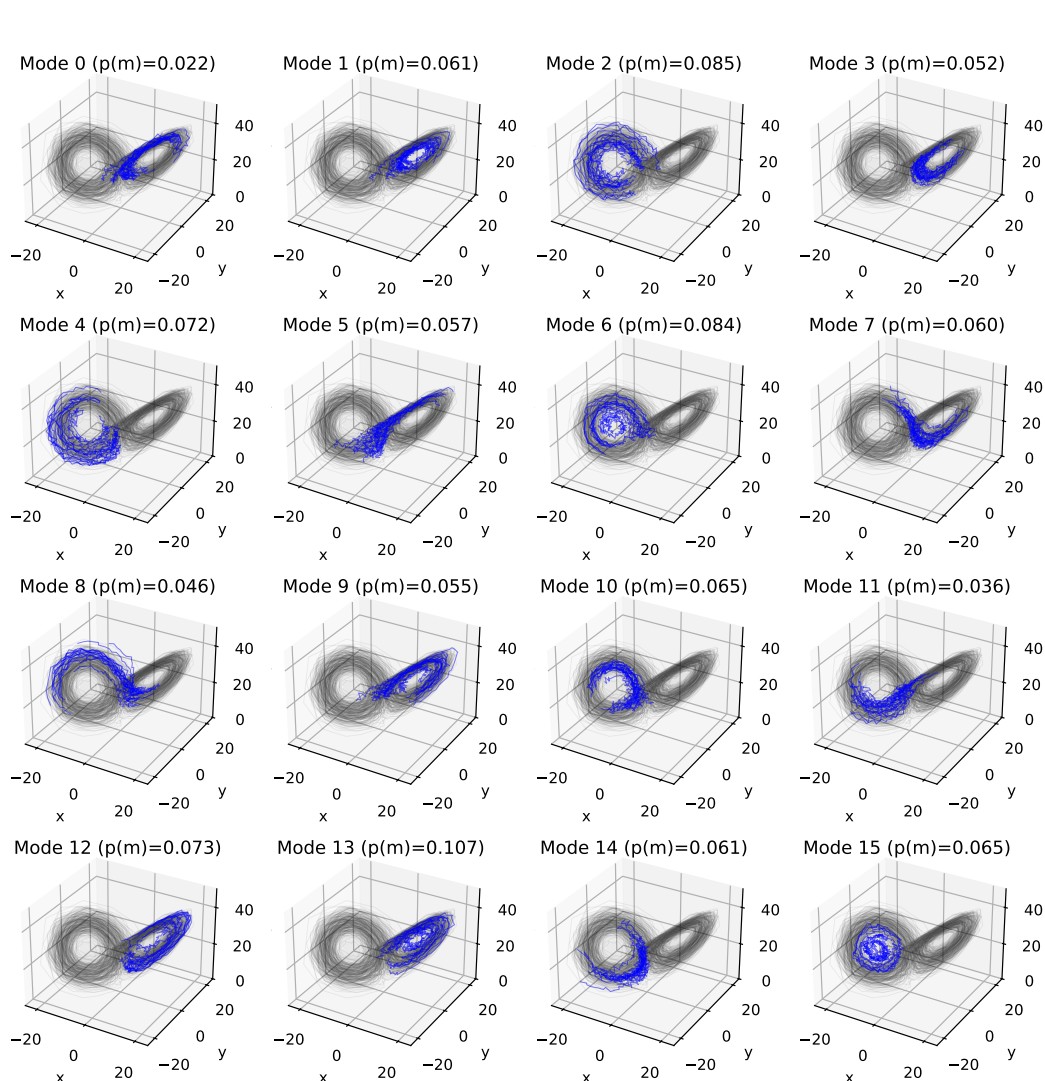

Figure 8: Mode-wise generation results on Lorenz63. Blue lines represent the generated trajectories from each mode.

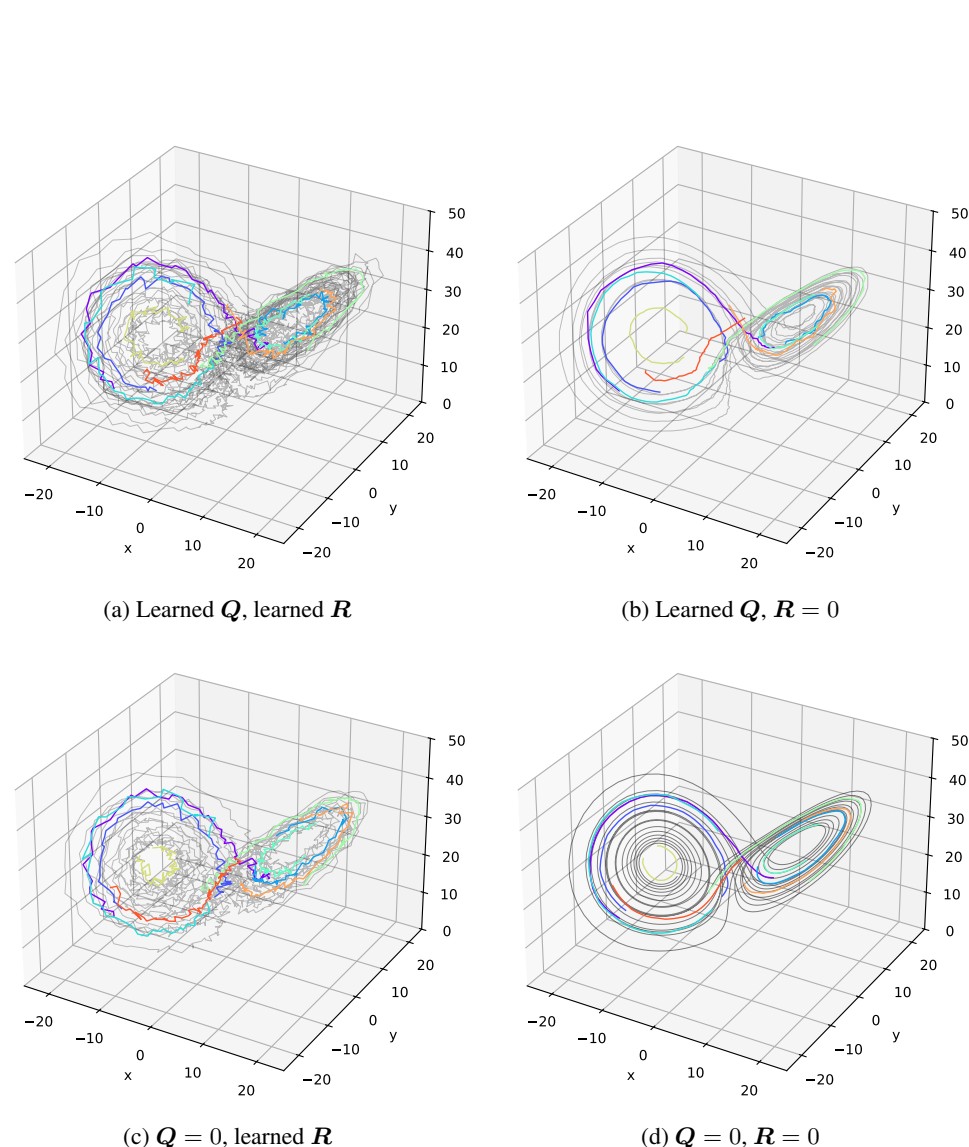

(a) Learned $\boldsymbol{Q}$, learned $\boldsymbol{R}$

(b) Learned $\boldsymbol{Q}$, $\boldsymbol{R} = 0$

(c) $\boldsymbol{Q} = 0$, learned $\boldsymbol{R}$

(d) $\boldsymbol{Q} = 0$, $\boldsymbol{R} = 0$

Figure 9: Denoising results on Lorenz63 using OUFlow. Colored lines represent generated trajectories by OUFlow. Black lines represent generated ground truth trajectories by solving Lorenz63 SDE under different noise conditions: (a) both system noise and observation noise present, (b) only system noise present, (c) only observation noise present, and (d) no noise.

