# OpenReview forum: "Fast Generation, Forecasting, and Imputation of Multivariate Irregular Time Series with OUFlow"
_ICLR.cc/2026/Conference — Submitted to ICLR 2026_

### Official Review · Reviewer_3BKs · 2025-10-29

**Soundness:** 3
**Presentation:** 3
**Contribution:** 2
**Rating:** 6
**Confidence:** 3

**Summary:**

This paper proposes a novel generative model OUFlow, focusing on three core tasks (generation, forecasting, and imputation) for multivariate, irregularly sampled time series data.

**Strengths:**

1. The paper has solid academic theory, with rigorous logic for the recommendation of the OU method, and the appendices supplement detailed formula derivations.
2. The experimental part is comprehensive, covering the three tasks mentioned in the paper as well as hyperparameter design.
3. The writing is well-organized, and all the figures follow a consistent, standard format.

**Weaknesses:**

1. Writing: What is "unconditional generation"? There are many formula derivations; although the proofs are sufficient, they seriously affect reading.
2. There is a slight disconnection between the Introduction section and the experimental section. The Introduction mainly emphasizes advantages in time complexity, but the experimental section contains only limited discussions on time complexity, and the results are not best.
3. Regarding the authors' statement that "making it suitable for broader applications such as anomaly detection", I did not find related discussions in the main text.
4. The paper has solid theoretical calculations, but its main goal is "Fast GENERATION". However, when look at the time complexity analysis and experimental results (Table 2, Table 3), the supposed "fast generation" advantage isn’t actually shown. What’s more, for time series tasks, the paper only uses a few datasets for comparison, which makes the experimental results less convincing.

**Questions:**

1. Table 1 shows that OUFlow does not show significant advantages in terms of time complexity. This can be observed in Table 1, Figure 3, Figure 4, and Figure 5, which is still a certain gap compared with the ACSSM method. The authors should provide a more sufficient explanation for this.
2. For the imputation task design, why is the random mask method not adopted, but instead a long time segment is selected for imputation? Meanwhile, the explanation given by the authors for the dataset shuffling method is somewhat far-fetched. These settings are different from those of previous imputation tasks.

---

> ### Author Response · Authors · 2025-11-28
> **Response to Reviewer 3BKs**
>
> Thank you for the constructive and helpful review. Your feedback made us realize that we did not clearly communicate the core selling point of our method, namely its generality. We have substantially revised the Abstract and Introduction and expanded the discussion of Further Applications to address this. Below we provide clarifications regarding the weaknesses you raised and responses to your questions.
>
> ### Weaknesses 1 \& 2
> > Writing: What is "unconditional generation"? There are many formula derivations; although the proofs are sufficient, they seriously affect reading.
> >
> > There is a slight disconnection between the Introduction section and the experimental section. The Introduction mainly emphasizes advantages in time complexity, but the experimental section contains only limited discussions on time complexity, and the results are not best.
>
> Thank you for the important feedback. We have revised the Introduction and Abstract to clarify the advantages of our approach and added concrete examples of tasks such as unconditional generation to improve clarity.
>
> ### Weakness 3
> > Regarding the authors' statement that "making it suitable for broader applications such as anomaly detection", I did not find related discussions in the main text.
>
> While anomaly detection can be implemented using standard likelihood-based procedures once the model’s likelihood is available, we had not detailed this in the main text. To substantiate the point within our generality claim, we added a Further Applications section discussing potential uses including anomaly detection, and we included small experiments on Lorenz63 in Appendix E.
>
> ### Weakness 4
> > The paper has solid theoretical calculations, but its main goal is "Fast GENERATION". However, when look at the time complexity analysis and experimental results (Table 2, Table 3), the supposed "fast generation" advantage isn’t actually shown.
>
> We apologize for the misleading emphasis. Our intended claim is that, among time-series generative models that support all three tasks for irregular time series, namely generation, forecasting, and imputation, our method exhibits favorable scalability with respect to the number of generated points. While ACSSM is indeed fast, it does not support generation and therefore does not contradict this claim. We have revised the wording in the Experiments section to consistently reflect this positioning.
>
> > What’s more, for time series tasks, the paper only uses a few datasets for comparison, which makes the experimental results less convincing.
>
> The four datasets we used were selected to span diverse properties across synthetic and real-world, periodic and non-periodic, and low-dimensional and high-dimensional settings. We therefore believe they adequately cover effectiveness across many practical cases.
>
> ### Question 1
> > Table 1 shows that OUFlow does not show significant advantages in terms of time complexity. This can be observed in Table 1, Figure 3, Figure 4, and Figure 5, which is still a certain gap compared with the ACSSM method. The authors should provide a more sufficient explanation for this.
>
> Thank you for highlighting this point. Our claim focuses on favorable scalability with respect to the number of generated points among general-purpose models supporting all three tasks. We do not claim superiority over ACSSM, which is specialized for forecasting and imputation and does not support generation. We made this distinction explicit in the Experiments section to avoid confusion.
>
> ### Question 2
> > For the imputation task design, why is the random mask method not adopted, but instead a long time segment is selected for imputation? Meanwhile, the explanation given by the authors for the dataset shuffling method is somewhat far-fetched. These settings are different from those of previous imputation tasks.
>
> Random-mask imputation is a valid protocol, but it often shortens missing intervals and can make interpolation easier, potentially compressing performance differences. We adopted long-duration imputation to present a more challenging scenario. Regarding the split strategy, using contiguous intervals for validation is known to shift the distribution between training and test in time-series settings; our approach aims to mitigate such shifts. That said, evaluation under alternative protocols (including random masks and extrapolative splits) is valuable, and we note this as an important direction for future work.

---

### Official Review · Reviewer_TNn6 · 2025-10-31

**Soundness:** 2
**Presentation:** 3
**Contribution:** 2
**Rating:** 4
**Confidence:** 3

**Summary:**

This paper presents OUFlow, a novel generative model designed for generation, forecasting, and imputation of multivariate irregular time series. The core innovation lies in integrating Ornstein–Uhlenbeck (OU) processes, linear observation, and normalizing flows to avoid the numerical integration steps required by traditional differential equation-based models. Experimental results on multiple datasets demonstrate that the proposed model outperforms existing methods in terms of both generation quality and computational efficiency.

**Strengths:**

1. The study addresses the challenging and highly relevant problem of generating multivariate regular time series—a topic of significant interest in both academic and applied contexts.
2. The model design exhibits a degree of novelty by combining the analytical properties of OU processes with the expressive power of normalizing flows, representing an interesting and promising direction.
3. The experimental section of the paper is quite good, covering the three core tasks of generation, forecasting, and imputation. The validation across multiple datasets demonstrates a substantial amount of work, which is commendable.

**Weaknesses:**

1. Although the paper claims OUFlow is a novel generative model, its core components (mixtures of OU processes, linear observation models, and normalizing flows_ are all established techniques. The primary contribution appears to be the combination and application of these methods to time series tasks, rather than the introduction of a fundamentally new theoretical or architectural breakthrough.
2. The paper states in Table 1 that OUFlow’s generation complexity is O (N+log(N+K)), repeatedly emphasizing its efficiency. However, the model’s theoretical time complexity is not optimal. Moreover, while Figure 5 compares generation time against the number of generated time points, it neglects the equally important impact of the number of observations on performance. Additional experiments are necessary to fully substantiate the claimed superiority in generation speed.
3. For experiments​:
a) While the selected baseline models (e.g., LatentSDE, DSPD) are reasonable, the paper overlooks several state-of-the-art generative models based on structured state space models (SSMs). Given the strong performance of recent SSM-based architectures (e.g., Mamba) in time-series modeling, comparisons on forecasting and imputation tasks should be included to ensure a comprehensive evaluation.
b) The original design intent and primary application scenarios of the ACSSM model are focused on forecasting and imputation tasks. The experimental results in this paper show that the performance of the OUFlow model is quite similar to ACSSM on forecasting and imputation tasks, yet demonstrates a significant performance gap on the generation task. It is recommended to supplement the comparisons with models specifically designed for generation tasks to enable a more equitable assessment of the model's comprehensive capabilities.
c) Evaluation criteria for generation tasks are not yet standardized across existing research, with different models employing varied methodologies, thereby reducing the comparability of results. For instance, the DSPD model utilizes a discriminative evaluation approach, where a classifier is trained and an accuracy rate of 50% is deemed the optimal indicator of generation quality. In contrast, this paper relies solely on the Mean Time-Averaged Energy Score (Mean TAES) as a single metric, failing to delve into the critical temporal properties of the generated sequences (such as spectral characteristics, long-term dependencies, or inter-variable correlations).
d) More mainstream datasets in the community should be used.
4. Section 3.5 notes that "in many cases, only a subset of modes is effectively learned, with the weights for all other modes approaching zero," yet the appendix D.3 suggests that increasing the number of modes improves performance. These statements appear contradictory. More detailed ablation studies are needed to clarify the influence of Mand the necessity of auxiliary loss functions.
5. About writing quality:
a) The abstract is overly brief and fails to adequately summarize the methodology and contributions.
b) The introduction begins with references to large language models and image generation, which are only loosely connected to the main topic of time-series analysis, weakening the focus.
c) The logic in the third paragraph is unclear and lacks emphasis, making it difficult to discern the key points.

**Questions:**

As in weaknesses.

---

> ### Author Response · Authors · 2025-11-28
> **Response to Reviewer TNn6 (1/3)**
>
> Thank you for several important observations. Based on your comments, we recognized that the Introduction and Abstract contained misleading statements, and we have revised them to clearly convey the primary advantage of our approach, namely its high generality. At the same time, some of the weaknesses you raised appear to stem from misunderstandings, so we provide clarifications and corrections below.
>
> ### Weakness 1
> > Although the paper claims OUFlow is a novel generative model, its core components (mixtures of OU processes, linear observation models, and normalizing flows_ are all established techniques. The primary contribution appears to be the combination and application of these methods to time series tasks, rather than the introduction of a fundamentally new theoretical or architectural breakthrough.
>
> These components do not function effectively by mere combination. First, the straightforward likelihood and posterior derivations scale cubically with the target-space dimension, making high-dimensional applications impractical; we derive formulations that scale linearly. Second, without careful parameterization of the OU process, computing matrix exponentials becomes necessary and can be a numerical bottleneck; we provide a general parameterization that avoids matrix exponentials and give closed-form transition probabilities. We also develop methodology and formulas that enable scan-style parallelization during generation. Furthermore, naive maximum-likelihood training is inadequate in practice; we propose a loss that substantially improves training stability and accuracy. We view these design choices and derivations—required to make the combination viable and tractable—as key contributions of the paper.
>
> ### Weakness 2
> > The paper states in Table 1 that OUFlow’s generation complexity is O (N+log(N+K)), repeatedly emphasizing its efficiency. However, the model’s theoretical time complexity is not optimal. Moreover, while Figure 5 compares generation time against the number of generated time points, it neglects the equally important impact of the number of observations on performance. Additional experiments are necessary to fully substantiate the claimed superiority in generation speed.
>
> We apologize that the title, abstract, and introduction may have led to a misunderstanding of our objectives. What we intended to emphasize is that, among time-series generative models that handle all three tasks for irregular time series, namely generation, forecasting, and imputation, our method exhibits favorable scalability with respect to the number of generated points. While ACSSM is indeed fast, it does not support generation and therefore does not contradict this claim. We have revised the Experiments section to reflect this positioning more precisely. Regarding the number of observations, our scaling is not particularly strong, and using “fast” broadly was misleading. Throughout the paper, we have replaced such phrasing with statements that specifically highlight scalability with respect to the number of generated points, and we explicitly add the limitation on observation-count scaling to the Conclusion as future work.

---

> ### Author Response · Authors · 2025-11-28
> **Response to Reviewer TNn6 (2/3)**
>
> ### Weakness 3
> > a) While the selected baseline models (e.g., LatentSDE, DSPD) are reasonable, the paper overlooks several state-of-the-art generative models based on structured state space models (SSMs). Given the strong performance of recent SSM-based architectures (e.g., Mamba) in time-series modeling, comparisons on forecasting and imputation tasks should be included to ensure a comprehensive evaluation.
>
> Mamba, cited as an example of a structured SSM model, is a neural network architecture rather than a generative model, so a direct comparison to OUFlow is not apples-to-apples. To the best of our knowledge, there is no structured-SSM-based general-purpose generative model that supports generation, forecasting, and imputation for irregular time series, which is our evaluation focus.
> That said, structured SSM models are known to deliver strong performance. We therefore built a probabilistic regression model using Mamba, specialized for forecasting and imputation only, and evaluated its accuracy. Concretely, we constructed a network that predicts the mean and variance for each component using Mamba, incorporated time embeddings to handle irregular sampling, set the number of parameters to more than twice that of OUFlow, trained it via maximum likelihood, and assessed accuracy using the same metrics as in our paper. The results show that OUFlow achieves higher accuracy on many tasks.
>
> #### TAES for forecasting
> |Model|Lorenz63|Exchange|Weather|Solar|
> |-----------|-----------|-----------|-----------|-----------|
> |Mamba-based model|$0.6988\pm 0.2570$|$0.2917\pm 0.1861$|$1.4931\pm 0.9621$|$\mathbf{3.1862\pm 2.9143}$|
> |OUFlow|$\mathbf{0.1783\pm 0.1464}$|$\mathbf{0.2222\pm 0.1447}$|$\mathbf{1.3721\pm 0.9943}$|$4.1373\pm 4.5752$|
>
> #### TAES for imputation
> |Model|Lorenz63|Exchange|Weather|Solar|
> |-----------|-----------|-----------|-----------|-----------|
> |Mamba-based model|$0.6038\pm 0.2283$|$0.2609\pm 0.1642$|$1.4600\pm 0.8131$|$\mathbf{2.8119\pm 2.4905}$|
> |OUFlow|$\mathbf{0.1005\pm 0.0231}$|$\mathbf{0.1491\pm 0.0897}$|$\mathbf{1.1238\pm 1.3190}$|$3.2993\pm 3.3743$|
>
> > b) The original design intent and primary application scenarios of the ACSSM model are focused on forecasting and imputation tasks. The experimental results in this paper show that the performance of the OUFlow model is quite similar to ACSSM on forecasting and imputation tasks, yet demonstrates a significant performance gap on the generation task. It is recommended to supplement the comparisons with models specifically designed for generation tasks to enable a more equitable assessment of the model's comprehensive capabilities.
>
> All four baselines we included in addition to ACSSM were chosen because they are well-suited for the generation task, specifically to assess generation alongside forecasting and imputation within the unified setting we target.
>
> > c) Evaluation criteria for generation tasks are not yet standardized across existing research, with different models employing varied methodologies, thereby reducing the comparability of results. For instance, the DSPD model utilizes a discriminative evaluation approach, where a classifier is trained and an accuracy rate of 50% is deemed the optimal indicator of generation quality. In contrast, this paper relies solely on the Mean Time-Averaged Energy Score (Mean TAES) as a single metric, failing to delve into the critical temporal properties of the generated sequences (such as spectral characteristics, long-term dependencies, or inter-variable correlations).
>
> TAES evaluates the agreement between the joint probability distribution across variables and time and the test data, thereby capturing multivariate time-series properties, including temporal and inter-variable correlations. It thus reflects spectral characteristics and long-term dependencies through the joint distribution. That said, depending on the application, targeted evaluations (e.g., spectral analyses or dependency-specific diagnostics) can be valuable; we consider this an avenue for future work.
>
> > d) More mainstream datasets in the community should be used.
>
> The datasets we use are commonly employed in time-series research and were selected to cover a broad spectrum (synthetic/real-world, periodic/non-periodic, low/high-dimensional). We therefore consider the selection appropriate for evaluating generality.

---

> ### Author Response · Authors · 2025-11-28
> **Response to Reviewer TNn6 (3/3)**
>
> ### Weakness 4
> > Section 3.5 notes that "in many cases, only a subset of modes is effectively learned, with the weights for all other modes approaching zero," yet the appendix D.3 suggests that increasing the number of modes improves performance. These statements appear contradictory. More detailed ablation studies are needed to clarify the influence of Mand the necessity of auxiliary loss functions.
>
> Appendix D.3 does not contradict Section 3.5; it supports the auxiliary loss’s necessity. As discussed, without the auxiliary loss, most weights collapse toward zero. Appendix D.3 reports results with the auxiliary loss introduced, which shows improved utilization of modes and better performance.
>
> ### Weakness 5
> > About writing quality: a) The abstract is overly brief and fails to adequately summarize the methodology and contributions. b) The introduction begins with references to large language models and image generation, which are only loosely connected to the main topic of time-series analysis, weakening the focus. c) The logic in the third paragraph is unclear and lacks emphasis, making it difficult to discern the key points.
>
> Thank you for the constructive suggestions. We recognized that the abstract and introduction obscured the key advantage of our approach, namely its generality, and we have revised these sections to foreground this point and improve focus and flow.

---

### Official Review · Reviewer_KBdz · 2025-10-31

**Soundness:** 3
**Presentation:** 2
**Contribution:** 3
**Rating:** 4
**Confidence:** 3

**Summary:**

The authors propose a time series modeling framework for forecasting, generation and imputation built on the idea of switching OU latent process which maps to observations through a normalizing flow, thereby defining an invertible transformation between the latents and the observations. This modeling choice allows the system to model irregularly spaced points and thus generalize to more real-world scenarios. The experiments indicate that it leads to improved performance as well as diverse generations, and in particular the choice of linear parameterizations leads to closed form evaluation of a number of relevant conditional distributions.

**Strengths:**

- The authors combine strong generative models (normalizing flows) with a latent OU process to model irregularly sampled time-series, pushing the frontier of real-world time-series modeling.
- The design choices allow computing of important conditional distributions tractable in closed form, thereby allowing ease of both sampling as well as likelihood computation.
- Empirically, they show that the proposed approach outperforms some of the baselines that they consider, built on top of state-space models as well as diffusion-style generative models.

**Weaknesses:**

- A potential weakness of the proposed method is that the choice of linear evolution of the latents is governed at the trajectory level as opposed to dynamically updated throughout the time-series, which feels quite limiting and non compositional.
- While the authors compare to some of the baselines, it would be nice to have a comparison against Time-Grad (Rasul et. al) which is one of the predominant methods for doing time series forecasting.
- Most of the quantitative results highlight performance as opposed to training and inference time, for both generation and forecasting/imputation tasks. It would be good to get a comparative estimate of the time required to train the proposed model (in contrast to baselines) as well as the computational cost at inference.

The authors should compare and contrast with the following existing works that look at time series forecasting using continuous time samplers as well as switching linear systems:

*Chen, Yu, et al. "Recurrent interpolants for probabilistic time series prediction." arXiv preprint arXiv:2409.11684 (2024).*

*Linderman, Scott W., et al. "Recurrent switching linear dynamical systems." arXiv preprint arXiv:1610.08466 (2016).*

*Halmos, Peter, Jonathan Pillow, and David A. Knowles. "System Identification for Continuous-time Linear Dynamical Systems." arXiv preprint arXiv:2308.11933 (2023).*

*Rasul, Kashif, et al. "Autoregressive denoising diffusion models for multivariate probabilistic time series forecasting." International conference on machine learning. PMLR, 2021.*

**Questions:**

- Maybe I did not understand this part correctly, but how do the authors parallelize training given the latents follow a linear dynamical system? What are the assumptions and simplifications made to allow for this parallelized training as opposed to ODE-RNN?

---

> ### Author Response · Authors · 2025-11-28
> **Response to Reviewer KBdz**
>
> Thank you for the insightful review that helped deepen the understanding of our approach. Below we provide clarifications and responses to the identified weaknesses and questions.
>
> ### Weakness 1
> > A potential weakness of the proposed method is that the choice of linear evolution of the latents is governed at the trajectory level as opposed to dynamically updated throughout the time-series, which feels quite limiting and non compositional.
>
> Thank you for the insightful perspective. We agree that using switching linear dynamics for latent variables is an effective way to increase expressive power. In OUFlow, instead of switching modes piecewise, we model complex distributions with a time-dependent normalizing flow to compensate for this. In practice, the ACSSM baseline employs a piecewise switching linear Gaussian model driven by observations, yet OUFlow matches or exceeds its predictive accuracy on many tasks.
>
> ### Weakness 2
> > While the authors compare to some of the baselines, it would be nice to have a comparison against Time-Grad (Rasul et. al) which is one of the predominant methods for doing time series forecasting.
>
> TimeGrad is a pioneering and representative diffusion model for time series; however, it targets only regular time series and is specialized for forecasting. Since OUFlow focuses on irregular time series, including TimeGrad in our benchmark would not be appropriate, and we therefore did not adopt it. That said, it is still interesting to assess how our general model performs against a specialized one. We thus compared OUFlow with TimeGrad on Lorenz63, where forecasting on regularly sampled data is feasible. The results show that OUFlow achieves performance comparable to TimeGrad.
>
> |Model|TAES for forecasting|
> |-----------|-----------|
> |TimeGrad|$0.1784\pm 0.1081$|
> |OUFlow|$0.1783\pm 0.1464$|
>
> ### Weakness 3
> > Most of the quantitative results highlight performance as opposed to training and inference time, for both generation and forecasting/imputation tasks. It would be good to get a comparative estimate of the time required to train the proposed model (in contrast to baselines) as well as the computational cost at inference.
>
> Our claim of “fast” pertains to inference only; we do not claim training-time advantages and do not report training-time metrics. In practice, OUFlow does not exhibit an advantage in training time compared with the other methods evaluated, and our intended use cases prioritize fast inference even when training is time-consuming.
>
> > The authors should compare and contrast with the following existing works that look at time series forecasting using continuous time samplers as well as switching linear systems:
>
> We appreciate the extensive pointers to prior work. Given the time constraints of the rebuttal period, we first report the comparison with TimeGrad as shown above. We will share results for the other suggested models once they are completed. That said, we emphasize that comparisons with task-specialized models are optional and not central to the paper’s main objective. We hope this context will be taken into consideration.
>
> ### Question 1
> > Maybe I did not understand this part correctly, but how do the authors parallelize training given the latents follow a linear dynamical system? What are the assumptions and simplifications made to allow for this parallelized training as opposed to ODE-RNN?
>
>  If by parallelization you refer to scan-style parallelization, this applies to generation/inference and is not used during training. OUFlow avoids the numerical integration required by ODE-RNN due to the analytical tractability of the OU process and the normalizing flow. Training proceeds without special parallelization assumptions beyond standard minibatching, and the tractable transition distributions allow us to compute the necessary likelihood terms directly rather than via ODE solvers.

---

### Official Review · Reviewer_9d8d · 2025-10-31

**Soundness:** 3
**Presentation:** 3
**Contribution:** 2
**Rating:** 6
**Confidence:** 3

**Summary:**

This paper introduces OUFlow, a unified framework that innovatively combines normalizing flows with Ornstein-Uhlenbeck SDEs for time series generation, forecasting, and imputation. Its key strengths include enabling exact likelihood computation, continuous-time modeling, and scalable inference by leveraging analytic solutions to SDEs, thus avoiding numerical solvers.

However, the work has significant limitations. The motivation is unclear, lacking a definitive problem statement or justification for why existing models are insufficient. The core assumption of a linear SDE process is potentially restrictive and not well-justified for complex, real-world dynamics. Empirically, the study is limited to a few clean, periodic datasets, raising concerns about generalizability to noisy or high-dimensional real-world data. Furthermore, it lacks a theoretical analysis of the model's expressiveness or convergence guarantees.

**Strengths:**

S1. The paper addresses time series generation, forecasting, and imputation under a unified framework, using appropriate metrics like energy distance and TAES, demonstrating a comprehensive and rigorous evaluation across multiple tasks.

S2. OUFlow creatively combines normalizing flows with Ornstein-Uhlenbeck-based SDEs, enabling exact likelihood computation and continuous-time modeling, offering a theoretically grounded alternative to discrete diffusion models.


S3. By leveraging analytic solutions to linear SDEs, the model avoids numerical solvers, ensuring stable training and scalable inference—particularly beneficial for long or irregularly sampled time series.

**Weaknesses:**

W1. The paper introduces OUFlow as a normalizing flow model for time series, but it fails to clearly define the core problem it aims to solve. The motivation is scattered. There is no discussion of why existing SDE-based or flow models are insufficient for the tasks, nor a compelling real-world use case that necessitates the proposed approach. This weakens the paper's narrative and perceived impact.

W2. The model assumes the latent process follows a linear SDE with block-diagonal drift matrix (Eq. 53). While this enables analytic solutions (Eq. 55), it is a strong and potentially restrictive assumption. Real-world time series often exhibit nonlinear, non-stationary, or high-dimensional dynamics. The paper does not justify why a linear OU process is sufficient or how the model would generalize to more complex dynamics. This raises concerns about expressiveness and modeling capacity.

W3. The experiments use only four datasets. While Lorenz63 is a classic chaotic system, the others are standard but relatively clean and periodic. There is no evaluation on high-dimensional, irregularly sampled, or real-world noisy data (e.g., medical, financial, or sensor data with missing values). This limits the generalizability and practical relevance of the results. Moreover, the absence of comparison to strong baselines weakens the empirical contribution.

W4. Despite being a flow-based model, the paper does not provide any theoretical analysis of the model’s expressiveness (e.g., universal approximation, density coverage), convergence, or consistency. For instance, under what conditions does the learned flow converge to the true data distribution? How does the OU prior affect the posterior? The derivation focuses on computational tricks (e.g., Woodbury) but lacks deeper theoretical insights that would elevate the work beyond an engineering contribution.

**Questions:**

Please see Weaknesses

---

> ### Author Response · Authors · 2025-11-28
> **Response to Reviewer 9d8d**
>
> Thank you for the helpful review. We agree that clarifying the problem setup is essential for establishing the effectiveness of our approach, and we have revised the Abstract and Introduction to make clear that our primary objective is to build a highly general model. On the other hand, several of the weaknesses raised appear to stem from misunderstandings; we therefore provide clarifications and corrections below.
>
> ### W1
> > The paper introduces OUFlow as a normalizing flow model for time series, but it fails to clearly define the core problem it aims to solve. The motivation is scattered. There is no discussion of why existing SDE-based or flow models are insufficient for the tasks, nor a compelling real-world use case that necessitates the proposed approach. This weakens the paper's narrative and perceived impact.
>
> We agree that the Introduction lacked clarity, and we have now made the motivation explicit. Our target problem is a unified treatment of irregularly sampled time series that supports generation, forecasting, and imputation with a single trained model. Existing works typically address subsets of these tasks or assume regular sampling; this creates friction when moving across tasks or dealing with irregular timestamps common in practice. We have revised the Introduction to foreground this unified objective.
>
> ### W2
> > The model assumes the latent process follows a linear SDE with block-diagonal drift matrix (Eq. 53). While this enables analytic solutions (Eq. 55), it is a strong and potentially restrictive assumption. Real-world time series often exhibit nonlinear, non-stationary, or high-dimensional dynamics. The paper does not justify why a linear OU process is sufficient or how the model would generalize to more complex dynamics. This raises concerns about expressiveness and modeling capacity.
>
> Although we assume a block-diagonal drift matrix, as stated immediately before these equations:
>
> > The eigenvectors $\mathbf{P}^m$ define the basis of the latent space, and their degrees of freedom can be absorbed by the redefinitions of $\mathbf{Q}^m$ and $\mathbf{H}^m$. Thus, fixing $\mathbf{P}^m$ to a specific matrix does not result in a loss of generality.
>
> this does not limit the model’s expressive power. OUFlow capture real-world high dimensionality via a linear observation model, and nonlinearity and nonstationarity via a time-dependent normalizing flow. Empirically, the high predictive accuracy across multiple tasks on nonlinear, high-dimensional datasets in the Experiments section supports the expressive power of OUFlow.
>
> ### W3
> > The experiments use only four datasets. While Lorenz63 is a classic chaotic system, the others are standard but relatively clean and periodic. There is no evaluation on high-dimensional, irregularly sampled, or real-world noisy data (e.g., medical, financial, or sensor data with missing values). This limits the generalizability and practical relevance of the results. Moreover, the absence of comparison to strong baselines weakens the empirical contribution.
>
> The Exchange dataset used in our experiments is a non-periodic, real-world financial dataset, and both the Weather and Solar datasets are noisy real-world datasets as well. While the original datasets are uniformly sampled, we split them into training and test partitions in a way that induces non-uniform sampling. As for baselines, DSPD is a diffusion model built on Transformers, and ACSSM combines Transformers with SSMs; notably, ACSSM achieved state-of-the-art across multiple tasks in 2025. Both are strong models.
>
> ### W4
> > Despite being a flow-based model, the paper does not provide any theoretical analysis of the model’s expressiveness (e.g., universal approximation, density coverage), convergence, or consistency. For instance, under what conditions does the learned flow converge to the true data distribution? How does the OU prior affect the posterior? The derivation focuses on computational tricks (e.g., Woodbury) but lacks deeper theoretical insights that would elevate the work beyond an engineering contribution.
>
> We acknowledge that our analysis is limited in areas such as expressivity. However, making the model viable required multiple mathematical techniques and considerations, which we have developed with careful theoretical rigor. For example, we derive the likelihood and posterior necessary to handle high-dimensional data by levaraging some matrices relationships. We also design OU process parameters to avoid matrix exponentials—which would otherwise be computationally expensive in a straightforward implementation—without loss of generality and, under this design, we provide explicit closed-form expressions for the transition probability distributions.

---

### Author Response · Authors · 2025-11-28
**General Response to All Reviewers**

We thank all reviewers for their constructive and helpful comments. Your feedback made us realize that the title, abstract, and introduction contained several statements that could be misread and may have caused confusion. While the title cannot be changed during rebuttal, we have revised the manuscript to clarify the key selling points of our approach and added a Further Applications section to support our claims of generality.

The primary contribution of our method is its high generality. There are very few time-series generative models that can handle irregularly sampled time-series data and support generation, forecasting, and imputation using a single trained model. Within this setting, our method demonstrates superior accuracy and favorable scalability with respect to the number of generated points among models that can perform all of these tasks. Moreover, due to its tractability, the model has potential applicability to additional tasks such as anomaly detection.

We acknowledge that we did not explicitly highlight the generality of the method, that we loosely described scalability with respect to the number of generated points as “fast,” and that we included ACSSM in the benchmark without sufficiently explaining that it is not a truly general-purpose model—all of which contributed to misunderstanding. In the revised manuscript, we (i) make the generality claim more explicit, (ii) provide clearer wording regarding scalability with respect to the number of generated points, and (iii) separate task-general models from task-specific ones in the benchmarking discussion to avoid conflation. We hope that these revisions will help reviewers develop a more accurate and deeper understanding of our approach.

Finally, we discovered an error in the normalization during inference in the OUFlow implementation. We have re-evaluated the experiments after correcting this issue and replaced the corresponding figures and tables in the Experiments section accordingly.

---

### Meta-Review · Area_Chair_11fk · 2026-01-06

**Summary:**

This paper proposes a unified framework for time series generation, forecasting, and imputation using normalizing flows with an Ornstein–Uhlenbeck SDE latent process. The topic is timely and the approach is promising, with empirical results that are competitive on the evaluated benchmarks. However, reviewers raised several concerns that are not fully resolved.

A key weakness is unclear positioning: the paper does not clearly define the core problem it aims to solve or explain why existing approaches are insufficient, and a reviewer states that the motivation remains somewhat vague. In addition, a reviewer points out that the reliance on a linear Ornstein–Uhlenbeck latent process with block-diagonal drift may be restrictive. Moreover, the experimental evaluation is also limited, with missing comparisons to state-of-the-art baselines for time-series generation. Generation quality is assessed using limited metrics, and the claimed efficiency advantage is not characterized.

The authors used the rebuttal to improve the manuscript, but given the large number of concerns, the submission would benefit from another round of revisions. I therefore recommend borderline reject, while encouraging resubmission.

**Reviewer Concerns:**

The paper needs to further improve the overall motivation, provide stronger justification of assumptions, and a more comprehensive evaluation.

**Reviewer Scores:**

I do not expect the discussion phase would have shifted the scores in either direction.

---

### Decision · Program_Chairs · 2026-01-26

Reject